# Value generalization in human avoidance learning

**Agnes Norbury[1]\*, Trevor W Robbins[2,3], Ben Seymour[1,4]**

[1]Computational and Biological Learning Laboratory, Department of Engineering, University of Cambridge, Cambridge, United Kingdom; [2]Department of Psychology, University of Cambridge, Cambridge, United Kingdom; [3]Behavioural and Clinical Neuroscience Institute, University of Cambridge, Cambridge, United Kingdom; [4]Center for Information and Neural Networks, National Institute of Information and Communications Technology, Suita City, Japan

**Abstract** Generalization during aversive decision-making allows us to avoid a broad range of potential threats following experience with a limited set of exemplars. However, over-generalization, resulting in excessive and inappropriate avoidance, has been implicated in a variety of psychological disorders. Here, we use reinforcement learning modelling to dissect out different contributions to the generalization of instrumental avoidance in two groups of human volunteers ($N$ = 26, $N$ = 482). We found that generalization of avoidance could be parsed into perceptual and value-based processes, and further, that value-based generalization could be subdivided into that relating to aversive and neutral feedback − with corresponding circuits including primary sensory cortex, anterior insula, amygdala and ventromedial prefrontal cortex. Further, generalization from aversive, but not neutral, feedback was associated with self-reported anxiety and intrusive thoughts. These results reveal a set of distinct mechanisms that mediate generalization in avoidance learning, and show how specific individual differences within them can yield anxiety.
DOI: https://doi.org/10.7554/eLife.34779.001

**\*For correspondence:**
aen31@cam.ac.uk

**Competing interests:** The authors declare that no competing interests exist.

## Introduction

During aversive decision-making, generalization allows application of direct experience with a limited subset of dangerous real-world stimuli to a much larger set of potentially related stimuli. For example, if eating a particular foraged fruit has led to food poisoning in the past, it may be adaptive to avoid similar-appearing fruit in the future. As an evolutionarily well-conserved process, generalization enables safe and efficient navigation of a complex and multidimensional world (*Sutton and Barto, 1998*; *Ghirlanda and Enquist, 2003*). However, *over*-generalization, resulting in inappropriate avoidance of safe stimuli, actions or contexts, has been suggested as a possible pathological mechanism in a range of psychological disorders including anxiety, chronic pain, and depression (*Duits et al., 2015*; *Dymond et al., 2015*; *Vlaeyen and Linton, 2012*; *Harvie et al., 2017*; *Pearson et al., 2015*).

Previous work on aversive generalization has focused on predicting punishments in passive (Pavlovian) designs. Such studies have revealed evidence of heightened subjective, physiological and neural responses to stimuli that bear perceptual similarity to learned exemplars (*Dymond et al., 2015*). However, the extent to which these observations extend to a decision-making context − that is whether or not to make an avoidance response in the face of certain stimuli, allowing us to exert *control* over experience of aversive outcomes − is unclear. Although Pavlovian processes can influence avoidance learning, the latter involves acquisition of a fundamentally distinct set of values relating to actions themselves. This is a clinically important distinction, as theories of many psychological disorders relate specifically to excessive avoidant behaviour over and above subjective fear

**eLife digest** People apply what they have learned from past experiences to similar situations, a phenomenon known as generalization. For example, if eating a particular food caused illness, a person will likely avoid foods that look or smell similar in the future. Generalization can be helpful because it allows people to decide how to act in new situations. But over-generalizing after a bad experience could lead an individual to fear benign scenarios. This may lead to unnecessary anxiety. It can also create a negative cycle where people avoid certain situations or objects, which prevents them from learning that they are safe.

Now, Norbury et al. show what happens in the brain when making decisions that involve generalization. In the experiments, volunteers were told seeing a particular flower design would lead to a painful electric shock, unless they pushed a button to 'avoid' that image. Individuals completed this task in a magnetic resonance imaging machine so Norbury et al. could observe their brain activity while they completed the task. A second group of individuals were asked to complete a similar task online, but instead of being shocked they lost money if they failed to hit a key when they saw the 'dangerous' flower. The online participants also filled out a survey about their experience of various psychological symptoms.

Norbury et al. used computer modeling to reconstruct how people decided whether or not to avoid images that looked similar to the harm-associated images but were in fact safe (did not lead to pain or losing money). The experiments showed that different parts of the brain were involved in different parts of the generalization process. Areas of the brain that interpret vision, fear, and safety played distinct roles. People who generalized more from harmful outcomes were more likely to report feeling anxious and having intrusive negative thoughts in their everyday lives. A better understanding of the brain processes that cause these symptoms in different situations might help scientists develop better treatments for conditions like anxiety in the future.

DOI: https://doi.org/10.7554/eLife.34779.002

(*Krypotos et al., 2015*) − for example, by reducing opportunities for extinction of inappropriate fear or allowing unnecessary avoidance to transfer to habit-based control (*Arnaudova et al., 2017*; *LeDoux et al., 2017*; *Gillan et al., 2014*).

There are a number of potential mechanisms by which avoidance generalization could be implemented by the brain. As emphasised in some accounts, perceptual uncertainty in stimulus identity alone can effectively yield generalization. Although there is debate about how well discriminative ability is controlled for in many generalization experiments (*Struyf et al., 2015*), there is good evidence that experience with aversive outcomes alters the representation of predictive stimuli in primary sensory cortices (*Weinberger, 2007*; *Sasaki et al., 2010*; *Wigestrand et al., 2017*), and that this may result in changes to absolute stimulus discriminability (*Resnik et al., 2011*; *Laufer and Paz, 2012*; *Aizenberg and Geffen, 2013*). On the other hand, generalization may also occur at the level of *value* representations, by the transfer of acquired value to similar, but discriminable cues during learning. In the Pavlovian case, several well-established behavioural phenomena implicate value-related processes at play in generalization across species (*Hanson, 1959*; *Schechtman et al., 2010*). That both perceptual and value processes might operate in parallel may explain why recent neuroimaging studies have highlighted different brain areas (e.g. limbic cortex vs primary sensory regions) as being key to Pavlovian aversive generalization in humans (*Onat and Büchel, 2015*; *Laufer et al., 2016*).

A further important factor in the control of avoidance learning is reinforcement by neutral (or 'safety') states, that signal omission of punishment. It is likely that generalization over these states can also influence behaviour: for example in the Pavlovian case, evidence for this is seen in 'peak-shift' effects, whereby the presence of a perceptually similar safety cue appears to inhibit response to nearby aversive cues (*Hanson, 1959*). It is therefore possible that under-generalization of safety cues, as opposed to over-generalization of aversive cues, might be a contributing factor to susceptibility to disorders such as generalized anxiety in humans (*Grupe and Nitschke, 2013*).

Here, we address three key questions: first, is there good evidence for generalization in avoidance learning in humans?; second, can we distinguish behavioural and neural components relating to

perceptual, aversive value, and safety value?; and third, which if any component predicts relevant psychological symptoms? We used a custom-designed perceptual task in conjunction with reinforcement learning modelling to study two groups: a laboratory-based sample (N = 26) who performed a pain avoidance task with concurrent neuroimaging (fMRI), and a larger cohort of individuals (N = 482), who performed a monetary loss avoidance task online alongside a battery of questionnaires designed to probe relevant psychological symptom dimensions (*Gillan and Daw, 2016*).

## Results

The overall study design is summarised in *Figure 1a*. In both groups of participants, generalization of instrumental responding was tested using a costly avoidance paradigm (*Figure 1c*). Briefly, participants were instructed that they would see a series of flower-like shapes on their screen, some of which were 'safe', and some of which were 'dangerous'. If they saw a dangerous shape and made no response, there was a high chance that they would receive a painful electric shock (fMRI sample), or lose 10 cents from their cash stake (online sample using Amazon Mechanical Turk, AMT). If they saw a safe shape, they would never receive a shock (or lose money) on that trial. In order to escape the possibility of a painful shock (or monetary loss) when they thought a dangerous shape had been presented, participants were told they could press the 'escape' button on their keypad. Participants were instructed that the aversive outcome would never occur on a trial when they had pressed the 'escape' button – but – that, importantly, pressing the button was associated with a small cost. Specifically, each time they pressed the escape button, it would be registered on a counter at the bottom of their screen. At the end of each block of the task, they would receive additional painful shocks (or lose additional cash) depending on how many times they had pressed the button during that block (one extra shock or 10 cent loss per every five button presses). The optimal strategy (in order to minimise the amount of pain received or money lost) would therefore be to press the button if they thought they saw a dangerous shape, but *not* press if they thought a safe shape was on the screen.

Crucially, on a small proportion of trials, the presented shapes were generalization stimuli (GSs). GSs were individually generated using precise estimates of perceptual ability (as measured on the first study session for the fMRI group) to be 75% reliably perceptually distinguishable from the task stimuli associated with aversive outcomes (CS+ s). (Due to time constraints and lack of control over testing environment, GS were generated based on average perceptual acuity from a pilot study in the online group.) The perceptual task (*Figure 1b*) was custom designed based on the recommendations of a recent review (*Struyf et al., 2015*). Specifically, in order to provide a fair test of perceptual performance during the generalization task, stimuli were not instantly comparable (in order to ensure that GSs would be reliably discriminable in an absolute sense, when presented in isolation; [*Slivinske and Hall, 1960*]), and testing occurred in the same emotional context (i.e. under threat of painful shock).

Importantly, the task stimulus array (in terms of arrangement of CS+ and CS- stimuli in perceptual space) was specifically chosen to probe asymmetries in generalization behaviour that result from value-based mechanisms – see *Figure 1b*. One such potential asymmetry is a characteristic shift in peak responding from the CS+ to surrounding GSs, away from the direction of the CS- in perceptual space (known as 'peak shift'), that has been proposed to result from the interaction of excitatory and inhibitory generalization gradients around CS+ and CS- stimuli following Pavlovian conditioning (*Hanson, 1959*). Crucially, the asymmetric array used here allowed us to compare responses to CS + GSs both near and far in perceptual space from the CS- – enabling detection of gradient interaction effects such as peak shift in instrumental avoidance, and allowing the separation of oppositely signed generalization gradients around CS+ and CS- stimuli.

We conducted a series of analyses on data from our two cohorts in order to address our key questions. First, we used reinforcement learning modelling to investigate whether there was evidence of value-based generalization in avoidance behaviour. Next, we used univariate fMRI data analysis to identify brain regions that encoded modelled internal quantities specific to value-based generalization processes. We then took a multivariate approach to investigate how the distributed representation of generalization stimuli in these regions changed over the course of the task, and how this related to individual differences in generalization. Finally, we used data from our online

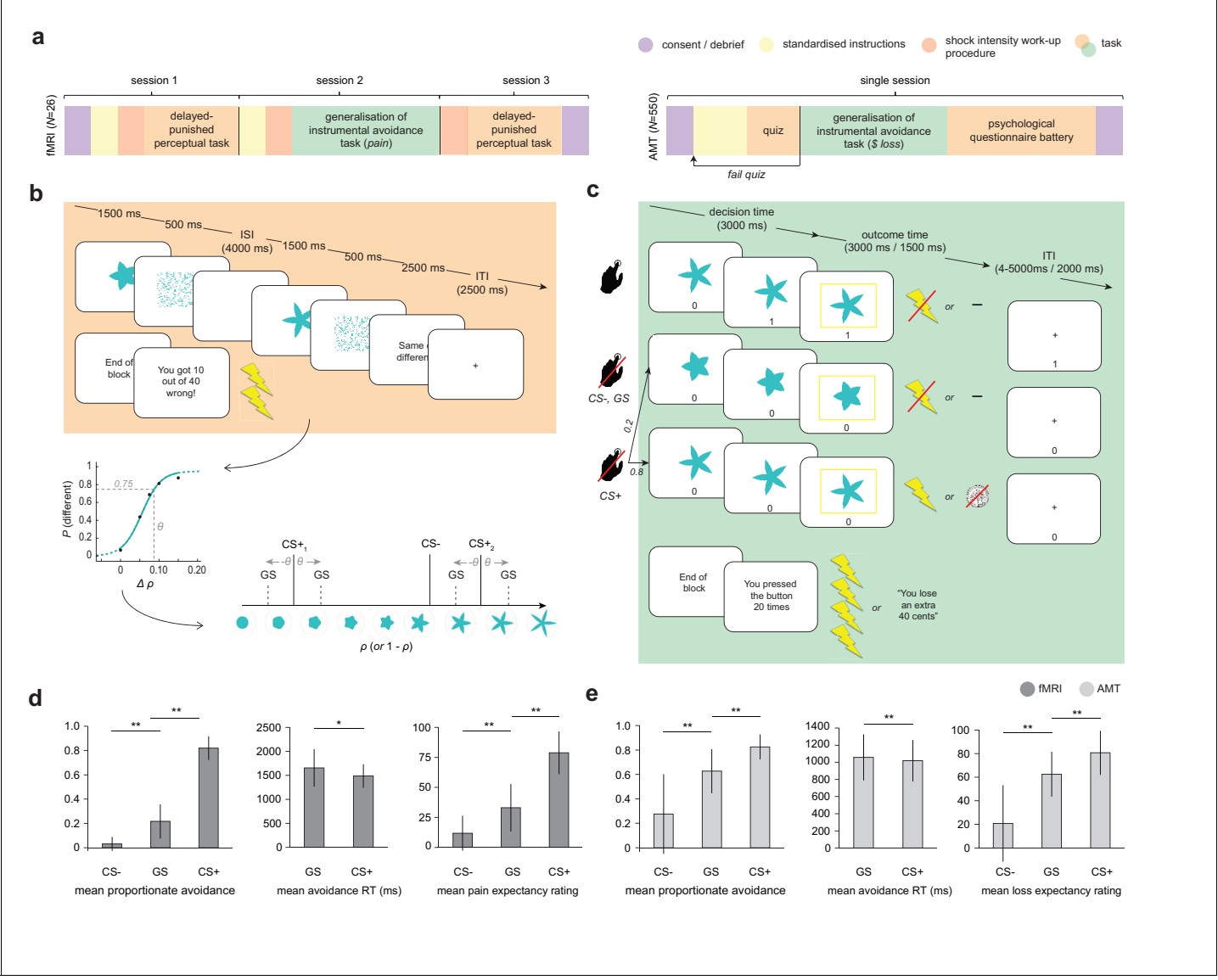

**Figure 1.** Study design and overall behaviour summary. (a) Study design and protocol for the two participant groups; fMRI, laboratory and functional imaging sample; AMT, Amazon Mechanical Turk (web-based) sample. (b) Delayed-punished perceptual task, used to determine 75% reliably perceptually distinguishable generalization stimuli (GSs) on in individual basis for the generalization of instrumental avoidance task (c) in the fMRI sample (in the AMT sample, GSs were generated based on mean perceptual acuity determined in pilot testing). (d) Summary of behaviour on the generalization task in fMRI and (e) AMT samples. ISI, inter-stimulus interval; ITI, inter-trial interval; CS+, conditioned stimulus with pain or loss outcome, CS-, conditioned stimulus with neutral outcome (no pain or loss). Error bars represent SD. *p=0.006, **p<0.001, paired sample *t*-tests.

DOI: https://doi.org/10.7554/eLife.34779.003

The following figure supplements are available for figure 1:

**Figure supplement 1.** Relationship between mean avoidance on generalization stimulus (GS) trials during the generalization of instrumental avoidance task, and mean post-task visual analogue scale pain/loss expectancy ratings.

DOI: https://doi.org/10.7554/eLife.34779.004

**Figure supplement 2.** Proportionate avoidance for individiual task stimuli (top row) and by CS type and block number (bottom row) for the generalization of instrumental avoidance task.

DOI: https://doi.org/10.7554/eLife.34779.005

**Figure supplement 3.** Effects of conditioning on perceptual acuity for task stimuli.

DOI: https://doi.org/10.7554/eLife.34779.006

questionnaire battery to determine whether specific elements of avoidance generalization were related self-reported psychological symptoms.

## Evidence for generalization in avoidance behaviour

For both groups of participants, the frequency of avoidance in response to generalization stimuli was intermediate to that evoked by CS- and CS+ stimuli (all p<0.0001, paired-sample $t$ tests; fMRI: GS $vs$ CS- $t_{25}$ = 7.57, mean difference = 0.18 [95%CI 0.14–0.24], GS $vs$ CS+ $t_{25}$ = −17.6, mean difference = −0.60 [95%CI −0.67 to −0.54]; AMT: GS $vs$ CS- $t_{481}$ = 27.0, mean difference = 0.35 [95%CI 0.33 – 0.38], GS $vs$ CS+ $t_{481}$ = −26.6, mean difference = −0.20 [95%CI −0.19 to −0.21]; *Figure 1d,e*). Despite never having been associated with the aversive outcome, participants also rated GSs significantly higher than CS- (but lower than CS+) stimuli on post-task pain/loss expectancy scales (all p<0.0001, paired-sample $t$ tests; fMRI: GS $vs$ CS- $t_{25}$ = 5.69, mean difference = 24.1 [95%CI 15–33], GS $vs$ CS+ $t_{25}$ = −8.14, mean difference = -52 [95% CI −39 to −66]; AMT: GS $vs$ CS- $t_{481}$ = 29.4, mean difference = 41.7 [95%CI 40.0–44.6], GS $vs$ CS+ $t_{481}$ = −16.5, mean different = -18 [95% CI −16.0 to −20.3], on visual analogue scales ranging 0–100; [*Figure 1d,e*]).

There was also a significant positive relationship between relative GS avoidance and relative GS pain/loss expectancy rating post-task in both groups (fMRI, Spearman's ρ = 0.655, p=0.00027; AMT, Spearman's ρ = 0.432, p=2.2e-16; both measures within-participant z-transformed, for relationships between raw scores see [*Figure 1—figure supplement 1*]). This suggests that a higher frequency of avoidance responding (plus associated lack of extinction) translated into higher conscious negative expectancy beliefs for generalization stimuli. There was no relationship between proportionate avoidance on GS trials and perceptual acuity at session 1 (individual θ values) or absolute intensity of the painful electrical stimulation (current amplitude) in the fMRI sample (all p>0.2).

This raises the question as to whether the observed avoidance on the GS trials was over and above that which would be expected from perceptual uncertainty alone. Notably, mean proportionate avoidance on GS trials in the fMRI group was around 0.2 (or ~0.25 when scaled relative to individual mean CS+ avoidance) – which, given that GSs were generated to be 75% reliably distinguishable from CS+s, is what might have been predicted from a purely perceptual account of task performance. Mean reaction times for making avoidance responses were also significantly slower for GS compared to CS+ stimuli in both groups, suggesting greater uncertainty on these trials (p=0.006, p=2.07e-11, paired sample $t$ tests; fMRI: $t_{25}$ = 3.00, mean difference = 167 ms [95% CI 51.2–282], AMT: $t_{481}$ = 6.87, mean difference = 38.8 ms [95% CI 27.7–49.9]; [*Figure 1d,e*]). To resolve this issue, we tested for the presence of additional value-based generalization processes in both datasets using a principled model comparison approach.

Simply, we fitted a series of reinforcement learning models to avoidance data from both samples (modified Q-learning algorithms, with trial-by-trial varying learning rates determined by the Pearce-Hall associability rule, [*Sutton and Barto, 1998*; *Le Pelley, 2004*] – see Materials and methods). Firstly, we fit a model with perceptual 'generalization' only (modelled as 25% chance of perceptual confusion between GSs and the adjacent CS+) – that is where all task stimuli were treated as independent states, with no transfer of value across states. Secondly, we fit a model with perceptual generalization plus an additional value-based generalization process. As there is some evidence that generalization functions are approximately Gaussian in shape, at least along a single perceptual dimension (*Ghirlanda and Enquist, 2003*), this was implemented as a Gaussian smoothing of stimulus value across perceptual space, with a single free parameter (σ) governing the width of this function. Thirdly, we fit a model with perceptual generalization plus two additional free parameters governing width of additional value-based generalization processes – one for aversive (shock/loss) and one for neutral (no shock/no loss) feedback ($\sigma_A$ and $\sigma_N$, respectively). This model was informed by previous empirical observations that generalization functions vary in gradient or width for aversive, neutral, and rewarding feedback (*Schechtman et al., 2010*; *Resnik and Paz, 2015*; *Laufer et al., 2016*).

The above models were fit to avoidance data from both groups using a variational Bayes approach to model inversion, under a mixed-effects framework (whereby within-subject priors are iteratively refined and matched to the inferred parent population distribution; see Materials and methods). Random-effects Bayesian model comparison indicated that in both samples the model with two additional value-generalization mechanisms (separately governing width of generalization from aversive and neutral feedback) best accounted for the avoidance data, as indexed

by exceedance probability (probability that the model in question was the most frequently utilised in the population; fMRI, EP = 0.823, AMT, EP =~ 1; *Figure 2a*).

For both fMRI and AMT data, this model provided a good account of avoidance decisions. Mean predictive accuracy ($r^2$, for binary choice data this is equivalent to the percentage of correct classifications) was 0.868 (±0.07) for fMRI and 0.849 (±0.11) for AMT groups, and the Bayesian 'p value' (posterior probability of the null hypothesis of random choice) was ≤6.8e-7 for all fMRI participants, and ≤0.026 for 477/482 AMT participants. In both groups, values of the parameter describing the width of aversive feedback ($\sigma_A$) were unrelated to values of other model parameters governing learning rate, choice bias, and choice stochasticity (see Materials and methods; all p>0.09), suggesting sufficient parameter identifiability. In both samples, $\sigma_A$ values were significantly larger than values of the parameter governing width of generalization from neutral (safe) feedback, $\sigma_N$, indicating wider generalization for aversive compared to neutral outcomes (p=3.0e-8, p=2.2e-16, related-samples Wilcoxon signed rank tests; fMRI: mean $\sigma_A$=0.752 ± 0.29, mean $\sigma_N$=0.028 ± 0.03; AMT: mean $\sigma_A$=0.695 ± 0.23, mean $\sigma_N$=0.057 ± 0.05). Interestingly, $\sigma_A$ values were not significantly related to $\sigma_N$ values (fMRI group, Spearman's $\rho$ = −0.169, p>0.4; AMT group, $\rho$ = 0.06, p>0.17), suggesting these may be at least partially independent processes.

Importantly, only a model including additional value-based generalization mechanisms can generate asymmetries in avoidance behaviour across pairs of generalization stimuli (peak shift), as apparent in *Figure 1—figure supplement 2*. Further, example traces for two representative participants from the fMRI group (*Figure 2b*) illustrate that stimulus values tend to asymptote – i.e. that under this model generalization of value across stimuli is assumed to be relatively constant over time. This

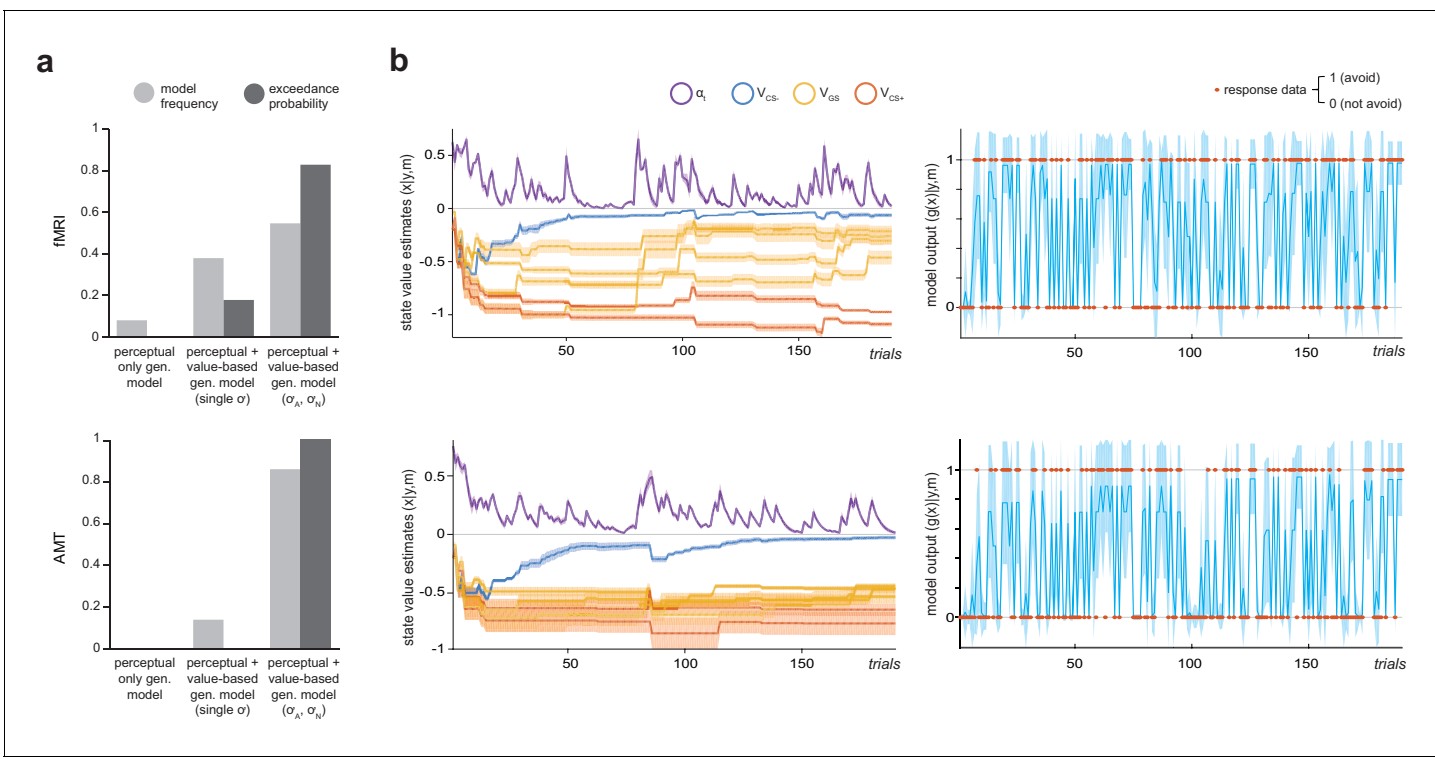

**Figure 2.** Computational modelling of instrumental avoidance behaviour. (a) Results of random-effects Bayesian model comparison for the laboratory (fMRI) and online (AMT) samples. For both groups, the best model was one that implemented both perceptual and additional value-based generalization between stimuli, with separate parameters governing width of generalization from aversive ($\sigma_A$) and neutral ($\sigma_N$) feedback. Model frequency, proportion of participants for whom a model was the best model; exceedance probability, probability that the model in question is the most frequently utilized in the population. (b) Ilustration of posterior state value estimates (x: the value of not avoiding for each CS, $V_{CS}$, plus the trial-varying learning rate, $\alpha_t$) and model output (g(x)) for the winning model (m) for a lower generalizing participant (top row) and higher generalizing participant (bottom row) from the fMRI group. Orange dots on the right hand side panels illustrate actual response data (y) on each trial. Shading represents variance of the posterior density.

DOI: https://doi.org/10.7554/eLife.34779.007

assumption is consistent with our behavioural data, in that a time-on-task analysis showed that after initial period of exploratory learning (blocks 1–2), generalization in terms of GS avoidance remains fairly stable. In both groups of participants, there were significant effects of both CS type and block number, and a CS type*block interaction, on proportionate avoidance responding (fMRI: $F_{2,50}$=406.3, $F_{4,100}$=6.14, $F_{8,200}$=8.68, respectively; AMT: $F_{2,962}$=1077.9, $F_{4,196}2$=24.3, $F_{8,3848}$=263.0, respectively; all p<0.001, repeated-measures ANOVA). In the fMRI sample, the CS type*block interaction was driven by lower avoidance for CS+ stimuli in block one compared to the rest of the task (p≤0.004; other CS types no significant differences between blocks; pairwise comparisons Bonferroni corrected for multiple comparisons). This suggests a strategy of exploratory non-avoidance to enable proper learning of CS+ stimuli in block 1, but fairly constant generalization of avoidance across later blocks. In the AMT sample, there was also lower avoidance for CS+ stimuli in block one vs other blocks (all p<0.001), but a decrease in avoidance for CS- stimuli in later blocks (3-5) vs earlier blocks (1 and 2; all p<0.001). Overall GS avoidance showed small increases then decreases over first three blocks (p<0.001), before stabilising between blocks 4 and 5 (p>0.5, Bonferroni-corrected pairwise comparisons; see [*Figure 1—figure supplement 2*]).

## Evidence for effects of conditioning on perceptual acuity

In the fMRI group, perceptual acuity for task stimuli was tested both before and after carrying our the generalization of instrumental avoidance paradigm, in order to test for possible effects of aversive conditioning on discriminability of the generalization stimuli (the three test sessions were carried out on three consecutive days for all participants, so any detected changes would likely reflect post-consolidation changes in perceptual performance).

There was no strong evidence for change in perceptual acuity in terms of θ value (difference in shape 'spikiness' parameter rho for 75% reliable perceptual discrimination) pre- *vs* post- conditioning (mean θ 0.071 ± 0.015 on session 1, 0.065 ± 0.019 on session 3; non-significant trend towards greater acuity on session 3, p=0.061, related-samples Wilcoxon signed rank test; [*Figure 1—figure supplement 3*]). Bayesian model comparison indicated that a model where generalization stimulus discriminability was held constant at 75% better accounted for avoidance data than one where discriminability was held constant at the estimated post-test (session 3) level, or a model where GS discriminability was assumed to be linear between session 1 and session three values (exceedance probability for the 75% constant model = ~1; [*Figure 1—figure supplement 3*]). Therefore GS discriminability was held constant across trials at 75% in all models.

## Differences in avoidance behaviour between lab-based and online cohorts

As can be seen in *Figure 1*, both mean avoidance and mean aversive outcome expectancy ratings for GSs (under non-avoidance) were higher in the AMT compared to the MRI sample (mean proportionate GS avoidance in MRI group: 0.22 ± 0.14, AMT: 0.63 ± 0.18; mean pain/loss expectancy rating [out of 100] in MRI group: 30 ± 23, AMT: 63 ± 19). One potential explanation for this difference is that there was lower absolute discriminability of generalization stimuli for the AMT participants. Although θ values (difference in ρ between CS+ and GS stimuli) were similar for the online and lab-based cohorts (0.071 ± 0.015 for the MRI group, and 0.065 for all AMT participants), we were unable to control factors such as participant distance from screen, and experimental window minimisation, that may have led to GSs being less discriminable than estimated in our pilot study (see Materials and methods). In addition, it is possible that participants conducting the study online paid less attention to the task than supervised lab-based participants (e.g. were multi-tasking), resulting in higher rates of stimulus-independent responding. Finally, it is possible that there were group-level differences in decision bias for the monetary loss compared to the pain reinforcer – for example due to differences in overall aversiveness between the two outcomes. Indeed, there was evidence of a difference in decision bias, as captured by the softmax bias parameter, between groups. The mean bias against deciding to avoid was 0.415 ± 0.14 in the MRI sample, and 0.315 ± 0.15 in AMT sample (p=0.0013, 95% CI for difference 0.04–0.16, $t_{28.5}$=3.56; Welch-Satterthwaite two-sample *t* test; *nb* large difference in *N* between groups).

## Brain regions encoding model quantities specific to value-based generalization

As our behavioural data provided evidence for the presence of generalization in instrumental avoidance in both groups, we next employed a univariate analysis approach to our functional imaging data in order to investigate whether model quantities specific to *value*-related generalization processes were encoded in regional blood oxygen level-dependent (BOLD) signals.

In addition to work highlighting the role of the insula, amygdala, and primary sensory cortex in aversive generalization following Pavlovian conditioning (*Ghosh and Chattarji, 2015*; *Onat and Büchel, 2015*; *Resnik and Paz, 2015*; *Laufer et al., 2016*), previous functional imaging studies have identified the striatum and prefrontal cortex as encoding generalization gradients in healthy human volunteers (*Dunsmoor et al., 2011*; *Greenberg et al., 2013*; *Lissek et al., 2014*). However, the contribution of perceptual uncertainty (i.e. absolute discriminability of 'generalization stimuli' compared with other conditioned stimuli) is not always adequately addressed in the study of such gradients. Here, we used a strict parametric approach to identify additional variance in regional BOLD that can be attributed to our winning value-based generalization model, *over and above* that which can be explained by a purely perceptual account. This was achieved by using serially orthogonalised regressors derived from each model to predict trial-by-trial variation in BOLD signal in our regions of interest (see *Figure 3a* and Materials and methods).

We found evidence for the encoding of additional variance in trial-by-trial expected stimulus values derived from the value-based generalization model in both the anterior insular cortex and the dorsal striatum (*Figure 3b*). BOLD signal was greater when the expected value of a particular stimulus was lower (or the predicted probability of receiving a painful shock if an avoidance response was not made was higher) in the left anterior insula ($p_{WB}$ = 0.0073, $k$ = 73, peak voxel [−30,23,−4], $Z$ = 4.71; sub-threshold trend in the right anterior insula: $p_{SVC}$ = 0.073, $k$ = 9, peak voxel [42,23,-1], $Z$ = 3.45), and right caudate ($p_{SVC}$ = 0.024, $k$ = 20, peak voxel [9,8,8], $Z$ = 3.95). There was no evidence for univariate encoding of this signal in primary visual cortex (V1) or the amygdala. We also found no evidence for *negative* encoding of aversive value (greater BOLD signal with lower predicted probability of shock, or 'safety signalling') in the ventromedial prefrontal cortex (vmPFC).

In addition to expected value signals, we examined potential encoding of prediction errors, which are the main learning signals in reinforcement learning (PEs; defined as the difference between actual and predicted outcome on any given trial – see Materials and methods). We focused our analysis on negatively signed PEs (generated on trials where no shock was received, but the predicted $P$(shock) was >0), as this both constrains analysis to trials where an avoidance response was not made (on avoidance trials PE = 0, by definition), and gives greater weighting to generalization trials where, due to perceptual uncertainty alone, predicted $P$(shock) will be >0, but no aversive outcome is ever delivered. (Positively signed PEs are highly collinear with shock administration and therefore are hard to detect under our design.)

We also found evidence of significant encoding of additional variance in PE signals from the value-based generalization model in insula and striatum (*Figure 3c*). Specifically, BOLD signal was greater when trial PE was more negative in the anterior insula, bilaterally (left: $p_{SVC}$ = 9.72e-5, $k$ = 93, peak voxel [−33,20,11], $Z$ = 5.48; right: $p_{SVC}$ = 0.024, $k$ = 19, peak voxel [33,26,-4], $Z$ = 4.35), right insula more posteriorly ($p_{SVC}$ = 5.85e-5, $k$ = 65, peak voxel [48,8,-4], $Z$ = 4.40), putamen, bilaterally (left: $p_{SVC}$ = 0.024, $k$ = 20, peak voxel [−27,−4,−1], $Z$ = 4.29; right: $p_{SVC}$ = 0.009, $k$ = 31, peak voxel [33,2,-1], $Z$ = 4.06), and right pallidum ($p_{SVC}$ = 0.046, $k$ = 14, peak voxel [18,5,2], $Z$ = 3.74). Significant clusters were also observed in the mid cingulate cortex ($p_{WB}$ = 0.001, $k$ = 103, peak voxel [6,14,44], $Z$ = 4.46), left parietal operculum ($p_{WB}$ = 3.56e-5, $k$ = 168, peak voxel [−48,−25,14], $Z$ = 4.10), right inferior parietal lobule ($p_{WB}$ = 0.003, $k$ = 90, peak voxel [54,-40,26], $Z$ = 3.82) and inferior frontal gyrus ($p_{WB}$ = 0.023, $k$ = 56, peak voxel [42,5,35], $Z$ = 4.31) − but we found no evidence of encoding of value generalization-derived PE signals in V1, the amygdala, or vmPFC.

## Changes in neural representation of generalization stimuli over the course of the task: relationship to individual differences in avoidance behaviour

Previous studies in animal models have shown that over the course of conditioning, the representation of the conditioned stimulus (CS+) in terms of response pattern across many individual units may

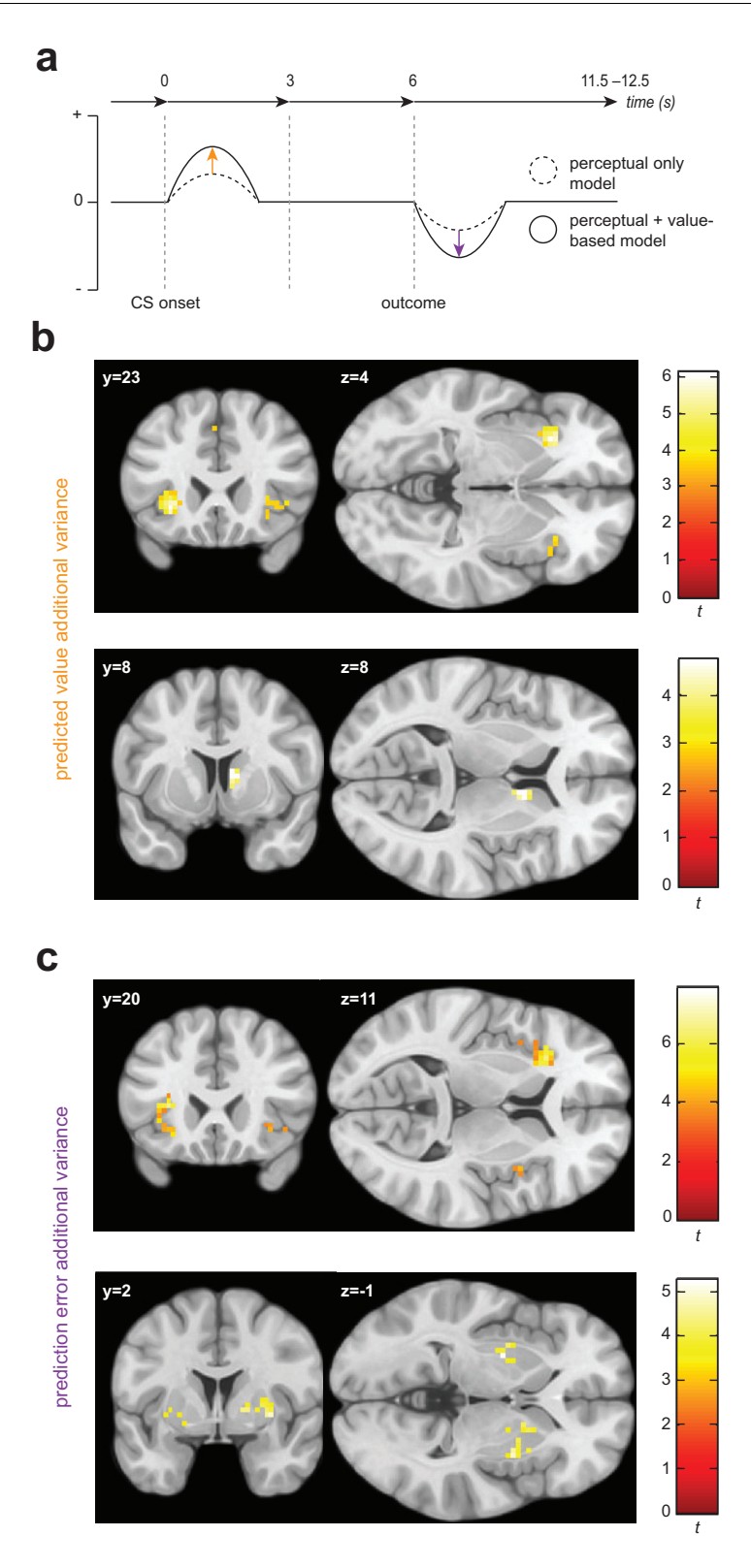

**Figure 3.** Univariate statistical maps highlight brain regions where changes in BOLD signal is significantly related to trial-by-trial variance in internal model quantities from the value-based generalization model, over and above that which can be explained by a purely perceptual account. (a) Schematic of a single trial for the fMRI group, showing the difference in estimated probability of receiving a shock (if no avoidance response is made) and outcome prediction error, as derived from the perceptual only vs the perceptual + additional value-based generalization models. (b) Significant

*Figure 3 continued on next page*

*Figure 3 continued*

encoding of additional value-based generalization in the expected value of each stimulus (likelihood of receiving a painful shock if no avoidance response is made), at the time of stimulus onset in the anterior insula and right caudate. (c) Significant encoding of additional value-based generalization as expressed in prediction error magnitude at the time of outcome receipt in the anterior insula, putamen, and right pallidum. Colour map shading represents *t* values.

DOI: https://doi.org/10.7554/eLife.34779.008

come to resemble that of the primary aversive reinforcer (e.g. *Grewe et al., 2017*). To complement our univariate results, we therefore examined how different task stimuli were represented in multivariate space using representational similarity analysis (*Kriegeskorte et al., 2008*). This approach enables the consideration of the full representational geometry across specific brain regions – *how* information is encoded, as well as whether or not it is – and depends on the calculation of distance metrics to quantify how (dis)similarly different kinds of stimuli are represented in multivariate space (in fMRI, across all voxels in a particular brain volume).

Following the approach of a recent study of aversive conditioning in rodents (*Grewe et al., 2017*), we examined how representational difference changed in our regions of interest earlier (blocks 1–2) vs later (blocks 3–5) in the task – and, crucially, how this change related to individual differences in overall behavioural expressions of conditioning. Specifically, we investigated whether changes in representation of GS, relative to CS+, stimuli over the course of the task related to individual tendency to generalize value from CS+ to GS stimuli – as captured behaviourally in avoidance responses on GS trials. We calculated a robust, cross-validated estimate of representational distance, Fisher's linear discriminant contrast (see Materials and methods, *Figure 4a*) in order to maximise the reliability of our results. Importantly, the use of a cross-validated distance measure means that derived (dis)-similarity estimates are unbiased by noise (which may potentially vary across individuals and imaging runs), and have a meaningful zero point (*Walther et al., 2016*).

Overall, for no region of interest was there a significant group level change in representational distance between GS and CS+ stimuli (all p>0.03, paired-sample *t* tests; Bonferroni-corrected threshold = 0.01 for alpha = 0.05). However, across individuals, greater increase in similarity of representation of GS to CS+ stimuli over the course of the task in primary visual cortex was related to greater behavioural generalization in terms of greater relative GS avoidance (p=0.010, multiple linear regression model; *Table 1*, *Figure 4b*). For individuals who made a higher relative proportion of avoidance responses towards generalization stimuli, V1 representation of GS stimuli came to be more similar to that of CS+ stimuli over the course of the task – but for individuals who avoided less on GS trials, GS stimuli came to be less similarly represented to CS+s in these regions (for visualisation of the relationship between raw proportionate GS avoidance and V1 distance change, see *Figure 4d*). There was no evidence of a significant relationship between GS−CS+ representational distance change and relative GS avoidance in the anterior insula, striatum, amygdala or vmPFC (*Table 1*, *Figure 4b*). We confirmed these results by implementing a cross-validated regularised regression (CV LASSO, see Materials and methods) on the same data (this kind of regression shrinks non-significant predictor coefficients to zero, and generally results in smaller coefficients compared to traditional linear regression). Under this robust approach, change in GS−CS+ similarity in V1, but not other regions, was retained as a significant predictor of relative GS avoidance (β = −0.040), in the model that minimised mean squared error (MSE).

Using a *post hoc* test, we examined whether changes in GS−CS+ representational distance in V1 might relate to changes in absolute discriminability of generalization stimuli (as measured on the day before and day after the generalization test session). Mean discriminability for GSs (CS+ ± θ) was 0.75 on session 1, by definition, and 0.79 on session 3 (±0.14, range 0.465–0.994; although note at the group level there was no significant change in θ values measured across sessions, see above). Under this exploratory analysis, we found evidence of a significant association between change in V1 GS−CS+ representational distance during the task, and post-conditioning changes in perceptual discriminability of the GSs. Individuals who showed an increase in similarity of representation showed worse perceptual performance post-(vs pre-) conditioning, and those who showed decreased similarity showing better performance (Spearman's ρ = 0.518, p=0.007; see [*Figure 4—figure supplement 1*]). There was no significant relationship between change in perceptual acuity and representational distance in any other brain region (all p>0.09).

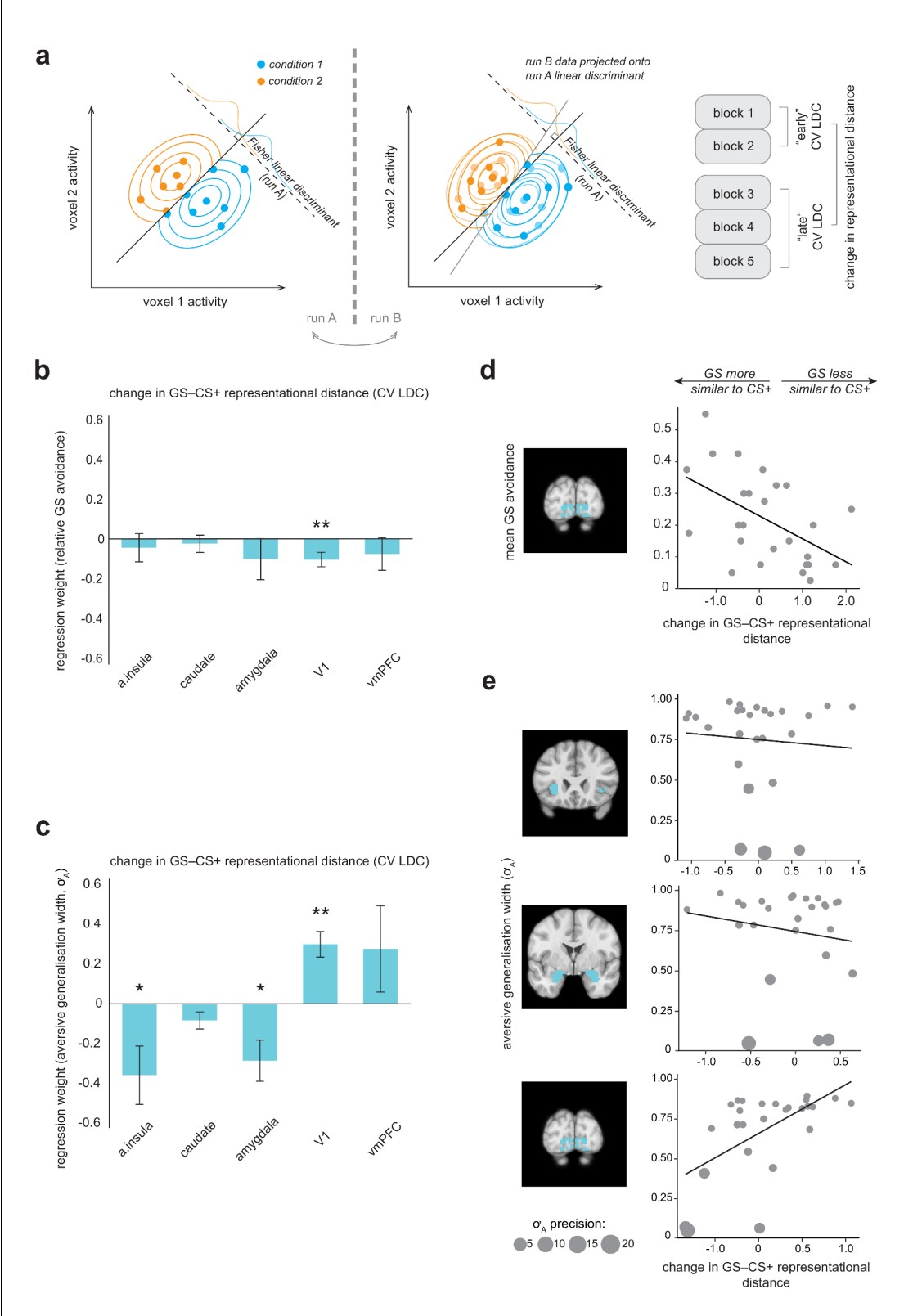

**Figure 4.** Multivariate fMRI results highlight regions where change in representational geometry over the course of the task between generalization stimuli (GSs) and pain-associated stimuli (CS+s) is related to individual differences in overall GS avoidance and the model parameter governing width of generalization from aversive feedback (σ_A). (a) Schematic of linear discriminant contrast analysis (based on [**Kriegeskorte et al., 2007**]). Within cross-validation folds, data from one imaging run is projected onto the optimal decision boundary derived from other runs, in order to remove inflation by
*Figure 4 continued on next page*

*Figure 4 continued*

noise in the final distance estimate (obtained by averaging across folds). (**b**) Multiple regression models detailing how changes in representational (dis) similarity over the course of the task in each ROI relate to overall relative avoidance on generalization trials, and (**c**) to individual differences in the model parameter governing width of generalization from aversive feedback. Error bars represent standard error. (**d**) Visualisation of bivariate relationships between change in representational geometry and raw GS avoidance (in primary visual cortex), and (**e**) between change in representational geometry and individual $\sigma_A$ values (in the anterior insula, amygdala, and V1), weighted by individual parameter estimate precision (1/ posterior variance). Larger bubble size represents greater precision (and therefore higher regression weight). Light blue shading on structural images illustrates the ROI volumes data were extracted from in each case. CV LDC, leave-one-out cross-validated linear discriminant contrast; a insula, anterior insula; vmPFC, ventromedial prefontal cortex. *p<0.05, **p<0.01.

DOI: https://doi.org/10.7554/eLife.34779.009

The following figure supplement is available for figure 4:

**Figure supplement 1.** Relationship between change in stimulus discriminability, pre vs post-conditioning, and change in GS−CS+ representational distance (CV LDC) in the primary visual cortex (V1) over the course of the generalization task.

DOI: https://doi.org/10.7554/eLife.34779.010

All the univariate fMRI findings presented above remained significant if re-ran using regressors derived from a model where perceptual discriminability of GSs changes linearly over the course of the task from pre- to post-conditioning measured acuity levels (full, unthresholded statistical maps for all analyses are available at Neurovault; neurovault.org/collections/3177).

## Changes in neural representation of generalization stimuli over the course of the task: relationship to individual differences in value-based generalization

We also sought to relate individual changes in similarity of representation of GS towards CS+ stimuli over the course of the task to individual model parameter estimates governing width of generalization, specifically from aversive feedback ($\sigma_A$ values).

We found that greater increases in similarity of representation of the GS relative to CS+ stimuli over the course of the task in the anterior insula and amygdala were related to larger generalization from aversive feedback parameter estimates (p=0.024, p=0.012, respectively, precision-weighted multiple linear regression model; see *Table 2*, *Figure 4c,e*). We also found that GS−CS+ representational distance change in V1 was related to individual differences in aversive feedback generalisation – in the opposite direction (p<0.001; *Table 2*). Somewhat counter-intuitively, increases in GS−CS+ similarity in V1 were associated with *lower* aversive value generalisation parameter values (*Figure 4c,e*). One possible explanation for this finding is that it is a result of V1-mediated changes in perceptual acuity for GSs – that is increased GS−CS+ representational similarity over the course of the task, associated with decreased perceptual acuity for GS stimuli, results in a lower requirement for additional value-based generalization in these individuals. Notably, this bi-directional relationship persisted if individual $\sigma_A$ values were re-calculated using a behavioural model that took into account potential conditioning-induced changes in perceptual acuity (i.e. perceptual discriminability of generalization stimuli changed linearly across trials from pre- to post- generalization test measured values; amygdala: $\beta = -0.353$, SE = 0.07, $t = -5.42$, p=2.65e-5; V1: $\beta = 0.204$, SE = 0.04, $t = 5.08$, p=5.77e-5). This suggests that a putative perceptual vs value-based generalization trade-

**Table 1.** Changes in representational distance (cross-validated LDC) with conditioning: relationship to overall generalization stimulus (GS) avoidance.

| Change in GS–CS+ representational distance | β | SE | t | p |
|---|---|---|---|---|
| a. insula | −0.04287 | 0.06798 | −0.631 | 0.535 |
| caudate | −0.02304 | 0.04173 | −0.552 | 0.587 |
| amygdala | −0.09792 | 0.09905 | −0.989 | 0.335 |
| V1 | −0.10072 | 0.03531 | −2.852 | 0.010* |
| vmPFC | −0.07407 | 0.07938 | −0.933 | 0.362 |

DOI: https://doi.org/10.7554/eLife.34779.011

**Table 2.** Changes in representational distance (cross-validated LDC) with conditioning: relationship to model parameter governing width of generalization from aversive feedback ($\sigma_A$). a. insula, anterior insula; vmPFC, ventromedial prefrontal cortex; V1, primary visual cortex; SE, standard error. *p<0.05

| Change in GS–CS+ representational distance | β | SE | t | p |
|---|---|---|---|---|
| a. insula | −0.357 | 0.146 | −2.448 | 0.024* |
| caudate | −0.082 | 0.043 | −1.908 | 0.071 |
| amygdala | −0.285 | 0.103 | −2.761 | 0.012* |
| V1 | 0.299 | 0.064 | 4.684 | <0.001* |
| vmPFC | 0.277 | 0.217 | 1.277 | 0.216 |

DOI: https://doi.org/10.7554/eLife.34779.012

off exists at the brain, rather than the behavioural level. Representational distance change in no region survived as a predictor of $\sigma_A$ values in the more robust CV LASSO model.

Although less well-studied compared to the aversive domain, there is evidence that the amygdala is also involved in the acquisition of information about *safety* in rodents and non-human primates (*Rogan et al., 2005*; *Genud-Gabai et al., 2013*), and that medial prefrontal entrainment of the amygdala is associated with learned safety (successful overcoming of generalized conditioned fear) in mice (*Likhtik et al., 2014*). This fits with a large literature on the vmPFC playing a role in 'safety signalling' in humans (*Fullana et al., 2016*). As a further exploratory analysis, we therefore investigated whether there was a relationship between change in GS-CS- similarity over the course of the task in the amygdala and vmPFC and individual values of the parameter governing width of generalization from neutral (non-pain) feedback, $\sigma_N$. (*Nb*, due to the arrangement of task stimuli, see *Figure 1b*, our design is not optimised to probe GS–CS- value generalization at the stimulus category level.)

We found evidence of significant relationships between GS−CS- similarity change in the amygdala and vmPFC and individual $\sigma_N$ values – such that individuals where representation of GSs came to be more similar to CS- in both these regions had greater neutral ('safety') generalization parameter values (amygdala: β = −0.043, SE 0.0086, t = −5.02, p=4.43e-5; vmPFC: β = −0.069, SE 0.009, t = −7.58, p=1.07e-7; precision-weighted multiple linear regression model). Representational change in the vmPFC (but not amygdala) was retained in the MSE-minimising CV LASSO model (β = −0.032).

## Relationship between individual differences in value-based generalization and self-reported psychopathology

Hypotheses about the role of generalization in psychological disorders tend to relate to an over-generalization of aversive information – but it has also been proposed that poor discrimination (e.g. between CS+ and CS- in anxiety groups) may be due to inadequate learning about safety cues. We therefore looked first at how psychological symptoms scores related to individual $\sigma_A$ values, but also examined possible relationships with individual $\sigma_N$ values, in our online cohort (N = 482).

Following the approach of *Gillan et al. (2016)*, the online group completed a battery of self-report questionnaires that probed symptoms hypothesized to be related to aversive over-generalization (trait anxiety, mood disorder symptoms, obsessive-compulsive traits, and 'global' cognitive style), in addition to some positive control measures (apathy and impulsivity scales). (A summary of scores on these measures and other demographic information for both samples is available in *Supplementary file 1*). To enable comparison with the findings of Gillan et al., self-report information was first compared to individual parameter estimates using precision-weighted linear regression models, controlling for age and gender identity (see Materials and methods). This approach was then complemented by the implementation of cross-validated regularised regression models (CV LASSO regression), as in the previous section (these models also included age and gender identity as regressors of no interest).

First, we sought to identify whether individual values of the parameter governing width of generalization from aversive feedback ($\sigma_A$) were related to symptom scores on any measure. Total scores

across measures exhibited good to excellent internal reliability (mean Cronbach's α = 0.882, see *Supplementary file 2*), and, as might be expected, covaried significantly across participants (mean absolute *r* for inter-correlation between scores = 0.479). Regression of total scores against parameter estimates was therefore implemented in separate models for each measure, in order to enable meaningful partition of variance. The Nyholt-Bonferroni corrected p value for significance across these separate models of non-independent measures was p<0.010 to maintain an alpha of 0.05 (effective number of independent variables = 5.0, see Materials and methods).

Parameter estimates governing width of generalization from aversive feedback were found to be significantly positively associated with trait anxiety scores (greater width with greater anxiety), and significantly negatively associated with trait apathy (smaller width with greater apathy; anxiety, p=0.009, apathy, p<0.001, individual precision-weighted linear regression models controlling for age and gender; see *Table 3*, *Figure 5a*). These two effects remained significant when trait anxiety and apathy scores were included in the same model, suggesting they were independent (anxiety: β = 0.050, SE 0.015, *t* = 3.34, apathy: β = −0.060, SE 0.014, *t* = −4.28; both p<0.001). This result was confirmed under the cross-validated and regularised analysis; when all predictors were entered in the same model both anxiety and apathy total scores were retained as predictors in the model that minimised MSE (β = 0.021, β = −0.032, respectively). No questionnaire total scores were significantly related to $\sigma_N$ values (p>0.05).

As per Gillan et al, we also sought to reduce collinearity in our battery of self-report measures by entering all recorded items (*N* = 142) into a factor analysis. Using an identical method to that described in the previously cited paper (see Materials and methods), we derived a three-factor solution (for scree plot see *Figure 5b*). These factors were labelled 'intrusive anxiety', 'low self-worth', and 'low self-control' on the basis of their top loading items (see *Figure 5c*).

The 'intrusive anxiety' factor was mostly composed of items from the trait scale of State-Trait Anxiety Inventory (STAI; 20 items, mean loading = 0.457 ± 0.12), Obsessive-Compulsive Index (OCI; 18 items, mainly items probing intrusive thoughts and checking behaviour, mean loading = 0.602 ± 0.087), Physician's Health Questionnaire (PHQ9; eight items probing mood disorder symptoms, mean loading = 0.531 ± 0.056), and the Barratt Impulsivity Scale (BIS; six items pertaining to racing/intrusive thoughts and restlessness, mean loading = 0.386 ± 0.15). 'Low self-worth' was mostly comprised of items from the Cognitive Style Questionnaire (CSQ; 37 items, mainly from low self-worth and internal attribution subscales, mean loading = 0.518 ± 0.13) and the STAI (11 items, mainly related to low self-worth/negative self-affect, mean loading = 0.322 ± 0.054). 'Low self-control' mostly comprised items from the BIS (23 items, mainly from the non-planning and attentional impulsivity subscales, mean loading = 0.485 ± 0.15), with some loading from the apathy motivation index (AMI; six items from the behavioural amotivation subscale, mean

**Table 3.** Relationship between width of generalisation from aversive feedback ($\sigma_A$ value estimates) and questionnaire total scores.

Each line represents the results of a separate model, as questionnaire scores were significantly collinear. STAI, Spielberger State-Trait Anxiety Inventory (trait scale); AMI, Apathy Motivation Index; OCI-R, Obsessive-Compulsive Index (Revised); PHQ9, Physician's Health Questionnaire 9 (a brief measure of mood disorder symptoms); BIS-11, Barratt Impulsivity Scale (version 11); CSQ global, Cognitive Style Questionnaire cognitive globalisation score. SE, standard error. *p<0.010 (Nyholt-Bonferroni corrected *p* value for multiple tests on non-independent data, alpha = 0.05).

| Questionnaire measure | β | SE | t | p |
|---|---|---|---|---|
| STAI total | 0.039 | 0.015 | 2.626 | 0.009* |
| AMI total | −0.051 | 0.014 | −3.687 | <0.001* |
| OCI-R total | 0.005 | 0.014 | 0.373 | 0.710 |
| PHQ9 total | 0.021 | 0.015 | 1.476 | 0.141 |
| BIS-11 total | −0.005 | 0.013 | −0.410 | 0.682 |
| CSQ global | −0.014 | 0.014 | −0.978 | 0.328 |

DOI: https://doi.org/10.7554/eLife.34779.013

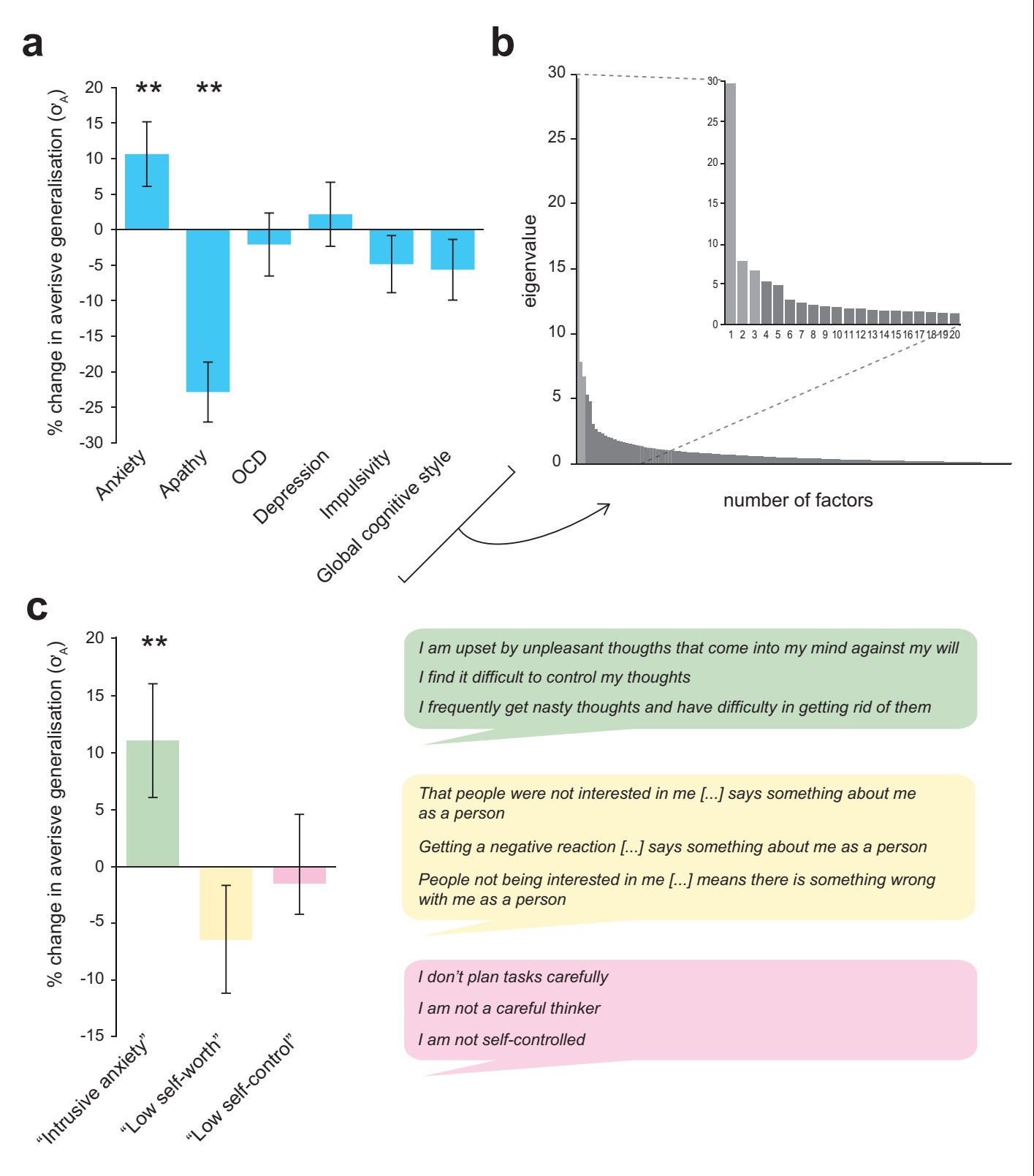

**Figure 5.** Associations between individual differences in aversive generalization and psychological symptom scores. (a) Percentage change in the model parameter governing width of generalization from aversive feedback ($\sigma_A$) with a one standard deviation increase in total score on each individual questionnaire measure used (individual regression models). (b) Scree plot indicating results of a factor analysis in which all response items from these measures ($N = 142$) were entered (inset, first 20 factors). A three-factor solution (lighter shaded bars) was indicated as the most parsimonious structure. *Figure 5 continued on next page*

*Figure 5 continued*

(c) Percentage change in $\sigma_A$ with an increase in 1 SD for each of the factor analysis-derived symptom scores (single regression model). The right hand panel shows the top three loading items for each factor, which were used to derive factor labels. Error bars represent standard error. **p≤0.009.
DOI: https://doi.org/10.7554/eLife.34779.014

loading = 0.356 ± 0.093) and STAI (seven items relating to feel uncontent/unrested, mean loading 0.321 ± 0.04). (For full item loadings for each factor, see *Supplementary file 3*).

The 'intrusive anxiety' factor analysis-derived symptom score was significantly and selectively related to individual differences in aversive generalisation width ($\sigma_A$ values) – in both multiple linear and robust regression models (p=0.008, precision-weighted multiple regression model; see *Table 4*, *Figure 5c*; only factor retained in MSE-minimising CV LASSO model, β = 0.019). None of the factor analysis-derived symptom scores were related to individual $\sigma_N$ values (all p>0.1).

## Discussion

The results presented here provide robust evidence for generalization in human avoidance learning. In particular, we demonstrate that generalization involves a number of distinct processes relating to different components of avoidance: perceptual uncertainty, aversive value generalization, and neutral (safety) value generalization. These processes each relate to different patterns of neural representations in the brain. Finally, we show that aversive value generalization is a specific predictor of trait anxiety in a large population sample.

Examining instrumental avoidance behaviour allows us to investigate how individuals learn about and attribute value to the set of *actions* they can take when faced with a particular stimulus or situation (as distinct from passively learnt Pavlovian stimulus-value associations). Using reinforcement learning modelling, we found behavioural evidence for additional value-based contributions to avoidance generalization (i.e. over and above that which might be expected from perceptual uncertainty alone) in two independent groups of participants (sampling different populations, and using two different kinds of aversive reinforcer). Notably, choice data from both groups supported an account of value-generalization that allowed for different widths of generalization from aversive (pain or monetary loss) vs neutral (no pain or loss) feedback. Consistent with previous evidence from studies of generalization of Pavlovian conditioning in humans and non-human primates, we observed larger width generalization functions for aversive compared to neutral feedback (*Schechtman et al., 2010*; *Resnik and Paz, 2015*; *Laufer et al., 2016*). In both groups, estimates of free parameters governing widths of these two processes were uncorrelated, suggesting they might relate to at least partially separable mechanisms.

Taking an explicit model-based approach enabled us to identify brain regions where BOLD signal was related to variance in modelled quantities specific to value-based generalization (namely, expected value and prediction error signals). When potential perceptual confusion between visually similar task stimuli was properly accounted for, we found evidence for encoding of value-related generalization signals in the anterior insula and dorsal striatum. The anterior insula and striatum (more ventrally) have previously been implicated in representing expected value and prediction error signals in higher-order pain conditioning (*Seymour et al., 2004*), and the dorsal striatum is implicated in prediction error signals in avoidance learning (*Palminteri et al., 2012*; *Seymour et al.,*

**Table 4.** Relationship between generalization width from aversive feedback ($\sigma_A$ value estimates) and factor analysis-derived symptom scores.
All factor scores were included in the same model. SE, standard error. *p<0.05

| Factor analysis-derived symptom score | β | SE | t | p |
|---|---|---|---|---|
| 'Intrusive anxiety' | 0.043 | 0.016 | 2.677 | 0.008* |
| 'Low self-worth' | −0.019 | 0.015 | −1.255 | 0.210 |
| 'Lack of self-control' | −0.000 | 0.014 | −0.032 | 0.975 |

DOI: https://doi.org/10.7554/eLife.34779.015

*2012*; *Eldar et al., 2016*), suggesting an important role for these structures in aversive learning (see also *Delgado et al., 2009*). Dorsal, rather than more ventral striatal control has also been implicated in the transfer from goal-directed to habit-based avoidance in instrumental paradigms (*LeDoux et al., 2017*). Greater understanding of habitual control in excessive avoidance has particular clinical relevance as it may explain why maladaptive avoidance can persist following extinction (e.g. contributing to treatment-resistance in exposure therapy for anxiety disorders, [*Treanor and Barry, 2017*]), and has been proposed as core mechanism in obsessive-compulsive disorder (*Gillan et al., 2014*). We found no evidence of univariate encoding specific to value-based model quantities in the amygdala, primary visual cortex (V1), or ventromedial prefrontal cortex (vmPFC). However, this may be because this kind of analysis is not ideally suited to detect distributed representations involved in associative learning.

In previous studies of Pavlovian aversive conditioning, it has been demonstrated that positively conditioned stimuli come to be more closely represented to the primary aversive outcome in multivariate space (e.g. across neural ensemble activity in the basolateral amygdala, [*Grewe et al., 2017*]). Here, we used a robust, cross-validated measure of representational distance to analyse data across all voxels in regions of interest, and found that increased similarity of representation of GS to CS +stimuli over the course of the task in primary sensory cortex was related to higher overall behavioural generalization (higher proportionate avoidance on generalization trials). Individuals for whom GS stimuli came to be more closely represented to CS +s in these brain regions (despite never having been directly associated with the aversive outcome) chose to avoid more in the face of GS stimuli − and vice versa. This change in representational geometry, in association with the lack of opportunity for extinction of inappropriately generalized value in an avoidance context, may have contributed towards the stability of generalization (in terms of overall GS avoidance) we observed over the later phases of the task.

Consistent with perceptual accounts of generalization, a post-hoc analysis suggested that representational change for GSs relative to CS +stimuli over the course of the task in primary visual cortex might account for some of the generalisation in avoidance we observed (in addition or parallel to value-based mechanisms identified above). Individuals who avoided more frequently on generalization trials, and who showed associated increases in GS−CS +representational similarity in V1, exhibited decreased perceptual acuity for task stimuli on next day perceptual testing - with the opposite pattern observed in participants who showed lower GS avoidance. Absolute decreases in discriminability for task stimuli result in increased generalization 'for free' (without having to involve additional mechanisms), and therefore may contribute to maintenance of generalization in some participants.

However, consistent with accounts that favour the involvement of a wider network of brain regions in coordinating generalization across stimuli, we also found a role for multivariate anterior insula and amygdalar representations in individual differences in aversive value generalization. Individuals who had higher estimates for the model parameter governing value generalization specifically from aversive feedback showed greater increases in similarity of GS−CS +representation in these regions. Somewhat surprisingly, the opposite relationship was observed in primary visual cortex, such that increases GS-CS +similarity in this region were associated with *lower* individual aversive generalization parameter estimates. One potential explanation for this finding is that some kind of compensatory mechanism exists between perceptual and value-based generalization processes, acting at the brain rather than behaviour level. Interestingly, changes in discriminative ability following aversive conditioning have recently been associated with altered insula and amygdalar processing of visual stimuli in humans (*Shalev et al., 2018*). However, this result was unexpected and would therefore benefit considerably from further investigation in future work.

Although less well optimised under our design, we also conducted an analysis to probe whether changes in GS relative to CS- stimuli might be associated with individual estimates of the model parameter governing width of generalization specifically from neutral (or 'safe') outcomes (in this case, omission of painful shock). Individuals with higher values of the parameter governing extent of generalization from neutral feedback exhibited greater increases in GS−CS- similarity over the course of the task in both the amygdala and vmPFC. This adds to a body of work suggesting that amygdalar function is not only important for the generalization of fear responses, but that it is also involved in safety learning (*Genud-Gabai et al., 2013*; *Likhtik et al., 2014*). A recent study in rodents suggests that the lateral amygdala may be particularly important region for understanding individual differences in fear behaviour towards perceptually ambiguous novel stimuli, with different

neuronal sub-populations involved in successful discrimination of novel safe stimuli and inappropriate fear responses – in a way that would be hard to detect by averaging signal across this region as a whole (*Grosso et al., 2018*). Although the vmPFC has previously been demonstrated to show inverse perceptual similarity-derived generalization gradients following aversive conditioning (e.g. *Lissek et al., 2014*; *Onat and Büchel, 2015*), it is not always clear from the experimental design whether this represents the simple inverse of aversive gradients (stemming from the CS+), or rather the positive signalling of safety gradients (stemming from the CS-). The evidence presented here provides tentative support for the latter account, at least in an instrumental context.

Excessive avoidance in response to contexts or stimuli which do not pose a threat to an individual's health or well-being can significantly impair general functioning and is often associated with high levels of psychological distress (*Arnaudova et al., 2017*). Such maladaptive avoidance has been identified as a core pathological dimension across several psychological disorders, including anxiety disorders, obsessive-compulsive disorder, chronic pain, and depression (*LeDoux et al., 2017*). Over-generalization of aversive feedback to encompass non-threatening but psychologically similar stimuli or contexts has been proposed as a key mechanism underlying the initiation and maintenance of excessive avoidance in these conditions (*Duits et al., 2015*; *Dymond et al., 2015*; *Harvie et al., 2017*; *Pearson et al., 2015*) – however, the link between generalization of negative value and inappropriate avoidance behaviour has been relatively underexplored.

We found selective relationships between psychological symptom scores and individual parameter estimates governing width of value generalization from aversive, but not safe/neutral outcomes. The largest positive relationship between symptom score and magnitude of aversive generalisation was for the factor-analysis derived score labelled 'intrusive anxiety', which mainly comprised items probing self-reported trait anxiety, but also reports of intrusive thoughts from the obsessive-compulsive inventory (% increase in parameter value with a 1SD increase in symptom score was 11.0% for intrusive anxiety, and 10.6% for trait anxiety alone). We also found a significant negative relationship between self-reported apathy and aversive generalization (22.9% decrease in parameter value with a 1SD increase in symptom score) – an effect which appeared to be independent from that relating to self-reported anxiety. This is an interesting finding, as we often think about apathy as involving a greater sensitivity to perceived effort, or decreased sensitivity to potential rewards, rather than a decreased impact of information about punishments (e.g. *Bonnelle et al., 2015*). As, to our knowledge, there has been no previously reported link between self-reported apathy and aversive generalization, this finding would benefit from future replication.

In summary, the findings reported here demonstrate the benefits of parsing complex processes such as generalization into separate components, and examining individual relationships between these components and both neural mechanisms and self-reported psychopathology. This approach may help unify previous apparently contradictory observations, and underlines that both perceptual and value-based processes are likely at work in generalization phenomena. Identification of patients across diagnostic categories who may have a primary deficit in excessive aversive generalization may help target them towards treatments which work more effectively. Further, greater understanding of the mechanisms of over-generalization of avoidance (including transfer to habit-based control systems) may help improve understanding of treatment resistance in these disorders.

## Materials and methods

### Code and data availability

All relevant code for stimulus generation, data collection, and data analysis, in addition to raw behavioural data, are available at the project's Open Science Framework page (osf.io/25t3f). Raw functional imaging data is deposited at openfMRI (openfmri.org/dataset/ds000249) and derived statistical maps are available at NeuroVault (neurovault.org/collections/3177).

### Design
#### fMRI sample
Protocol

Each participant completed three testing sessions on 3 consecutive days. On the first day, participants were pre-screened, gave informed consent, and performed initial sensory acuity testing for

the generalization task stimuli. On the second day, participants completed the generalization of instrumental avoidance task (performed in fMRI scanner, using individually-generated conditioning stimuli [CSs] derived from day 1perceptual performance), followed by visual analogue scale (VAS) ratings of pain expectancy for each CS. On the third day, participants repeated the perceptual acuity test.

All participants were recruited via online advertisement. Exclusion criteria were left-handedness and history of neurological or psychological illness, in addition to usual MR safety criteria. The sample size was chosen on the basis of a power calculation. Previous functional imaging studies in humans have found effect sizes in the region of r = ~0.5 for generalization-related BOLD signal and individual difference measures (*Greenberg et al., 2013*; *Lissek et al., 2014*; *Cha et al., 2014*). We calculated that a sample of N = 26 would allow us to detect r = 0.5 with an alpha of 0.05 and power of 80%, two-tailed (correlation point biserial model, G*Power version 3.1.9.2). Volunteers were paid £20/hr in recompense for their time and discomfort arising from the painful electrical stimulation. The study was approved by the University of Cambridge Psychology Research Ethics Committee.

## Delayed-punished perceptual discrimination task

Prior to starting the task, participants were introduced to the shock and electrode and a work-up procedure was performed (as described below) to set the level of painful stimulation. The delayed-punished perceptual task was then carried out, as summarized in *Figure 1b*. Briefly, on each trial, participants viewed an individual shape (target or comparison stimulus, order randomized on each trial), followed by a mask (scrambled mean shape image), delay period (blank screen), second shape, and second mask. At the end of each trial, participants had to indicate whether they thought the two shapes had been the same, or different. The inter-stimulus delay period of four seconds was chosen to be long enough such that comparison of stimuli could not be achieved by instantaneous mechanisms, but required comparison in short-term memory (e.g. primate data suggests discrimination performance for visual features decreases significantly from <1 s to around 4–5 s inter-stimulus delay, [*Pasternak and Greenlee, 2005*]), and roughly matched to the inter-trial interval from the generalization task. There were 16 trials per absolute value interval per target (160 trials total), and trials were divided into four equal blocks. At the end of each block, participants received feedback on how many incorrect judgments they had made, and received a proportionate number of painful electric shocks as punishment (one painful shock per five incorrect judgments).

Stimuli were five-fold radially symmetric flower-like shapes, as described in *van Dam and Ernst, 2015*. These were selected on the basis that they can be continuously generated along a single perceptual axis of 'spikiness' using the mathematical description provided in the paper, and psychophysical evidence demonstrating that they are perceptually linear (i.e., that discrimination thresholds are constant along this axis). Shape 'spikiness' is parameterized by a single value, $\rho$ (where $0 < \rho < 1$), which relates the inner and outer radii of the shape such that stimuli are of constant surface area. Target stimuli were shapes with $\rho$ values of the two CS+ stimuli from the generalization task (0.25 and 0.75). These target stimuli were compared to comparison stimuli of intervals of ±0, 0.05, 0.075, 0.1, and 0.15 $\rho$, such that the possible range of different shapes was well tiled. Participants worked on a pre-defined set of comparison stimuli (opposed to a stair-cased approach) so that pre-exposure to conditioning task stimuli (and therefore opportunity for perceptual learning) would be matched across individuals.

## Generalization of instrumental avoidance task (pain version)

Participants completed five blocks of 38 trials each. On each trial, participants were presented with a stimulus in the centre of their screen. This initiated a 3 s decision period, during which they must decide whether or not to make an 'escape' (avoidance) response. Following this, a yellow bounding box appeared around the shape, indicating the time when an avoidance response could be made was over and they would receive the outcome for that trial. If an avoidance response was made, no shock was ever delivered on that trial. If no avoidance was made, and the stimulus was a 'safe' shape (CS-), no shock was delivered. If the stimulus was a 'dangerous' shape (CS+), a painful shock was delivered on 80% of non-avoidance trials at the end of this outcome period (i.e. 6 s from stimulus onset, *Figure 1c*).

On a low frequency of trials, shapes were generalization stimuli (GSs; 2 presentations of each GS per 38 trial block). These stimuli were individually generated to be 75% reliably distinguishable from

adjacent CS+ s based on day 1 perceptual task performance (see *Figure 1b*), and were never associated with painful shock. Trial types were presented in the following ratio: 10 CS-: 10 CS+(*2): 2 GS (*two per CS+) in a pseudorandom sequence, in order to minimise learning about GS stimuli. Although previous studies have tended to employ designs with multiple generalization stimuli, use of a single GS around each CS+ in perceptual space is the most efficient design if the perceptual discriminability of probe stimuli is accurately known, and you are agnostic as to the precise identity of the generalization function (e.g. exponential vs Gaussian, assuming this constant across individuals). Frequency of individual GS presentation (10 per GS) was comparable to recent functional imaging studies of Pavlovian generalization (e.g. 7 and 34 presentations per GS during generalization test phases, respectively: [*Laufer et al, 2016*; *Onat and Büchel, 2015*]).

The stimulus array was asymmetric in perceptual space (see *Figure 1b*), with two CS+ (and four associated GS) stimuli – one nearer and further from an intermediary CS-. This array was chosen in order to probe the presence of characteristic asymmetries in conditioned responses that are hypothesised to arise from the interaction of oppositely signed generalization gradients (e.g. peak shift, [*Hanson, 1959*]), and on the basis of previous observations that change in perceptual discriminability of aversively conditioned stimuli (CS+ s) may depend on the relative 'nearness' of safety stimuli (CS-s) in perceptual space (*Aizenberg and Geffen, 2013*). Axis direction (in terms of increasing or decreasing 'spikiness') was counterbalanced across participants.

## Online sample
### Protocol
In order to test relationship with real-world psychological symptoms in an appropriately powered sample, an online version of the study was also carried out, following the approach of Gillan et al. (*Gillan and Daw, 2016*; *Gillan et al., 2016*). Participants were Amazon Mechanical Turk (AMT) workers based in the USA (in practice, had an AMT account linked to US bank with provision of an US social security number). Participants were required to be over 18 years of age, but otherwise remained anonymous.

Participants completed an online consent procedure, and provided limited demographic information (age and gender identity). They then read several screens of detailed task instructions (including visual examples of sample trials), based on the standardized instructions given to lab study participants. Participants were required to pass a 10 item true/false quiz on task structure before continuing (scoring less than 10/10 returned participants to the instruction screens). They then performed a monetary loss-based version of the generalization of instrumental avoidance task (see below), followed by a battery of questionnaires probing psychological symptoms and cognitive style.

We calculated that a final sample size >459 should be powered to detect a small effect size of 0.13 or greater (association between behavioural and self-report parameters), at alpha = 0.05 and 80% power (two-tailed point biserial model). As expected attrition following quality control was ~15% (*Gillan et al., 2016*), we collected $N$ = 550 complete datasets, yielding a final expected sample size of ~468.

Payment rates were based on UK ethical standards for online experiments (equivalent to a minimum of £5 ph). Participants were paid a flat rate of $2.50 for taking part, plus up to around $3.00 additional bonus payment depending on task performance. The average bonus payment was $2.21 (±0.82) and the average time between accepting and submitting the task was 42 min (equivalent to $6.72 mean hourly payment rate). The study was approved by the University of Cambridge Psychology Research Ethics Committee.

### Generalization of instrumental avoidance task (loss version)
The generalization task was identical in structure to that performed by the lab-based participants, but used monetary loss instead of painful shock as the aversive reinforcer (*Figure 1c*). Prior to starting the task, participants were endowed with a $6.00 stake, and instructed that, although a certain amount of loss was inevitable, whatever total remained at the end of the task would be paid directly to them as a bonus (the loss therefore had real-world value). As BOLD data was not being collected, trials were slightly shorter than for the fMRI group (second set of timing figures, *Figure 1c*) – although the length of the decision period was kept the same.

Perceptual testing was not performed in the online sample due to time constraints, and the inability to control the testing environment (e.g. participant distance from screen, window size, etc.) over the course of testing. Generalization stimuli were therefore the same for all participants, and generated on the basis of mean perceptual performance on the perceptual task in a pilot sample. This pilot testing was carried out under the same conditions and timing parameters as described for the MRI sample, with the exception that no punishment shocks were administered (and no pain-delivery apparatus was attached to participants).

## Questionnaire battery

Following completion of the generalization task, participants completed several self-report measures (questionnaire order was randomized across participants). These measures were chosen to probe psychological constructs hypothesized to be related to over-generalization of aversive outcomes (anxiety, depression, and obsessive-compulsive symptomatology), as well as positive controls that might suggest a more general effect of psychopathology on task performance (impulsivity, apathy). Questionnaires consisted of the trait scale of the State-Trait Anxiety Inventory (STAI; [*Spielberger et al., 1970*]); the Physician's Healthy Questionnaire 9 (PHQ9; [*Martin et al., 2006*]), a brief measure of mood disorder symptoms; the revised (short-form) Obsessive-Compulsive Index (OCI-R; [*Foa et al., 2002*]); the Barratt Impulsiveness Scale v11 (BIS-11; [*Patton et al., 1995*]); and the Apathy Motivation Index (AMI; [*Ang et al., 2017*]). All chosen measures have previously been shown to be suitable for use in the general population.

A short version of the Cognitive Style Questionnaire (CSQ-SF; [*Meins et al., 2012*]) was also administered. This self-report measure asks participants to imagine themselves in various scenarios (e.g. 'Imagine you go to a party and people are not interested in you'), and then probes the imagined causes of this scenario along dimensions of 'internal', 'global', and 'stable' attributions, plus low self-worth. On this measure, a more 'global' cognitive style reflects a tendency to attribute negative events to causes which are general, rather than specific (a cognitive form of over-generalization), and has been found to be a predictor of future depressed mood (*Pearson et al., 2015*). The CSQ-SF was administered at the end of the battery of questionnaires for all participants in order to avoid possible mood-induction effects.

## Quality control procedure

Following previous studies utilizing AMT (*Crump et al., 2013*; *Gillan et al., 2016*), a number of exclusion criteria were applied sequentially to the dataset to attempt to exclude poor quality responses. Firstly, we excluded participants who made avoidance responses on less than 50% of total CS+ trials (indicating lack of learning/random responding on these trials), $N = 62$. Secondly, we further excluded participants who selected the wrong answer to a catch item inserted into the questionnaire battery ('Please select the answer 'a little' if you are reading this question'), $N = 6$. 68 datasets were excluded in total (12.3% of those collected), yielding a final sample size of 482. Questionnaire data quality was further assessed via calculation of internal reliability coefficients for each measure (Cronbach's $\alpha$).

# Data collection

## fMRI sample

Stimulus presentation and response collection was coded using Cogent2000 v1.30, run in Matlab R2015b (Mathworks). Perceptual testing on day one and three took place in a laboratory, and generalization testing in an fMRI scanner. Size of stimuli in terms of visual angle subtended were matched between lab and scanner environments in order to ensure ~constant discriminability.

For the painful stimulation, electric current was generated using DS7A constant current stimulator (Digitimer), delivered to a custom fMRI-compatible annular electrode (which delivers a highly unpleasant, pin-prick like sensation), worn on the back of the participant's dominant (right) hand. All participants underwent a standardized intensity work-up procedure at the start of each testing day, in order to match subjective pain levels across sessions to a level that was reported to be painful, but bearable (8 out of 10 on a VAS ranging from *0* ['no pain'] to *10* ['worst imaginable pain']). The pain delivery setup was identical for lab-based and MR sessions.

Functional imaging data were collected on a 3T Siemens Magnetom Skyra (Siemens Healthcare), equipped with a 32-channel head coil. Respiration data were collected during functional scanning

using a pneumatic breathing belt (BrainProducts), and choice (avoidance) data were recorded using an MR-compatible button box.

Field maps were acquired in order to correct for inhomogeneities in the static magnetic field (short TE = 5.19 ms, long TE = 7.56 ms, 32 × 3 mm slices). Five functional sessions of 212 volumes were collected using a gradient echo planar imaging (EPI) sequence (TR = 2000 ms, TE = 30 ms, flip angle = 90°, tilt=-30°, slices per volume = 25, voxel size 3 × 3 × 3 mm; this included three dummy volumes, in addition to the three pre-discarded by the scanner). Limited field of view (constrained by equipment used for additional physiological data collection) was aligned to the base of brain and angled away from the orbits, such that there was full coverage of the occipital and temporal lobes, plus prefrontal cortex. A T1-weighted MPRAGE structural scan (voxel size 1 × 1 × 1 mm) was also collected. Full sequence metadata are available at openfMRI (openfmri.org/dataset/ds000249).

## Online sample

The experiment was coded in javascript using jsPsych ([*de Leeuw, 2015*]; available at github.com/jspsych/jsPsych), and was deployed to Amazon Mechanical Turk via the psiTurk engine ([*Gureckis et al., 2016*]; available at github.com/NYUCCL/psiTurk). The experiment was hosted in the cloud using an Amazon Web Services EC2 instance. A more detailed description of this setup is available at osf.io/mjgtr. The task was not made available on mobile devices (phones or tablets) in an attempt to ensure minimum screen size.

## Analysis

### Perceptual acuity

For fMRI sample participants, psychometric functions (a logistic function with free parameters governing slope, bias, and lapse, or stimulus-independent error, rate) were fitted to response data from the perceptual task using the psignifit toolbox v2.5.6 (available at bootstrap-software.org/psignifit), run in Matlab. Formally,

$$P(\text{diff}) = 1/(1 + \exp((\alpha - \Delta\rho)/\beta))$$

where $P(\text{diff})$ is the probability of reporting the comparison shape as different (restricted between the bounds of 0 and 1−lapse rate), $\Delta\rho$ is the difference in shape parameter $\rho$ between target and comparison stimuli, and $\alpha$ determines the bias, and $\beta$ governs slope, of the logistic function. This toolbox implements the constrained maximum-likelihood method of psychometric function fitting described in *Wichmann and Hill (2001)*.

Individual psychometric functions were then used to calculate the different in $\rho$ value necessary for the comparison stimulus to be distinguishable from the target on 75% of trials (henceforth, $\theta$).

### Instrumental avoidance behaviour

Avoidance behaviour was modelled using a set of modified Q-learning algorithms (*Sutton and Barto, 1998*). Each stimulus was modelled as a different state, with the value of executing each action (*avoid* or *notAvoid*) in each state ($V_{s,a}$) updated after each trial ($t$) on the basis of a simple Rescorla-Wagner rule – that is, on the basis of difference between the predicted value of that state-action pair, and the actual outcome of each trial ($R_t$; coded as 0 for no shock/no loss and −1 for shock/monetary loss). Formally,

$V_{s,a,t+1} = V_{s,a,t} + \kappa^*\alpha_t{}^*(R_t - V_{s,a,t})$

Learning rate ($\alpha_t$) was updated on each trial, according to the empirically well-supported Pearce-Hall associability rule (*Le Pelley, 2004*):

$\alpha_{t+1} = \eta^*|(R_t - V_{s,a,t})| + (1 - \eta)^* \alpha_t$

According to this rule, the learning rate on each trial is determined by the absolute magnitude of past prediction errors, such that state-action value estimates are updated by more when previous outcomes have been more surprising, and by less when they were less surprising. This allows for learning in terms of modelled value adjustment to be greater when outcomes are more surprising (e.g. at the start of the task), but to be lesser (leading to more stable values) when outcomes are better predicted. A non-constant learning rate also ensures that parameters governing width of value-based generalization, which are assumed to be constant over the course of the task, are identifiable during parameter estimation (see below equations). Individual differences in degree of

dependence on prediction error history and overall scaling of learning rate are governed by the free parameters $\kappa$ and $\eta$.

To model perceptual 'generalization' (possibility of identity confusion between GSs and adjacent CS+ s), the value of not avoiding for GSs on any given trial was defined as:

$V_{GS,notAvoid} = 0.75 * V_{GS,notAvoid,t} + 0.25 * V_{adjacent\ CS+,notAvoid,t}$

For the models with additional value-based generalization, on each trial the values of all states were updated in proportion to their perceptual similarity to the current state, $i$, using a rule similar to those employed in previous studies (*Kahnt et al., 2012*; *van Dam and Ernst, 2015*) – that is according to a variable-width Gaussian function across perceptual space. For each state, $j$:

$G_j = 1/\exp((\rho_i - \rho_j)^2 / (2 * \sigma^2))$

$V_{j,a,t+1} = V_{j,a,t} + \kappa * \alpha_t * (R_t - V_{i,a,t}) * G_j$

where $\rho$ is the parameter governing shape 'spikiness', and the width of Gaussian function governing generalization is determined by the free parameter $\sigma$. For the fMRI sample, average $\rho$ values were used during model fitting for all subjects, as stimuli had been matched in subjective perceptual space. For the 2-width model, different $\sigma$ values were fit depending on whether the outcome for that trial was aversive or neutral ($\sigma_A$ and $\sigma_N$, respectively).

As participants were explicitly instructed that they would never receive the aversive outcome if they made an avoidance response, the value of avoiding in any state ($V_{s,Avoid,t}$) was held constant at 0. Value estimates were fit to binary choice data via a softmax observation function, taking into account the cost of making an avoidance response (additional shock or unit monetary loss to be received at the end of that block for every five button presses made, or 0.2 per avoidance decision):

$P(\text{avoid}) = 1/(1 + \exp(-\beta * (V_{s,avoid,t} - V_{s,notAvoid,t} - 0.2 - \text{bias})))$

where the free parameter $\beta$ determines how driven $P$(avoid) is by the difference in value between the two possible actions ($V_{s,avoid,t} - V_{s,notAvoid,t}$), and the *bias* parameter determines overall bias towards choosing a particular action (avoiding or not avoiding).

For both samples, models were fit to choice (avoidance) data using the variational Bayes approach to model inversion implemented in the VBA toolbox ([*Daunizeau et al., 2014*]; available at mbb-team.github.io/VBA-toolbox), run in Matlab. Model fit was performed in a mixed-effects framework. Simply, after the first round of model inversion, the individual posterior free parameter value estimates are used to approximate the population distribution these values were drawn from, which is then used as prior for the next round of inference, until convergence (no further group-level reduction free energy). This approach reduces the likelihood of outliers in any individual parameter estimates.

Model comparison was by random-effects Bayesian model comparison (*Rigoux et al., 2014*). This method of model comparison assumes that the population is composed of subjects that differ in terms of the model that describes them best, then induces a hierarchical probabilistic model that can be inverted to derive the posterior density over model frequencies, given participants' data. Under this approach, the critical metric for any given model is its exceedance probability, or the likelihood that that particular model is more frequent than all other models in the comparison set.

## Functional imaging data

### Pre-processing

Functional imaging data were pre-processed using SPM12 (Wellcome Trust Centre for Neuroimaging, www.fil.ion.ucl.ac.uk/spm) in Matlab. Briefly, functional images were realigned to the first functional image in each sequence, unwarped, corrected for time of acquisition, and normalized to MNI space via tissue probability maps derived from the co-registered structural image. The full pre-processing pipeline available is available at osf.io/f9drs as a BIDS-compatible Matlab script (*Gorgolewski et al., 2016*). Finally, images were smoothed via convolution with an 8 mm full-width at half-maximum Gaussian kernel for the univariate (but not multivariate) analysis.

Breathing belt data were processed using the PhysIO toolbox ([*Kasper et al., 2017*]; available at translationalneuromodeling.org/tapas), which provides physiological noise correction for functional imaging data using the Fourier expansion of respiratory phase implemented in the RETROICOR algorithm (*Glover et al., 2000*).

## Univariate analysis

Functional imaging data were first analysed according to a mass univariate approach based on the general linear model for time series data in each voxel, as implemented in SPM12. This enables detection of whether variance in BOLD in each voxel is significantly related to modelled internal quantities (i.e. if particular model terms are encoded in BOLD signal time course), with relative spatial specificity. Several models were fit to individual BOLD time series data using restricted maximum likelihood estimation to produce individual statistical maps at the first level, which were used to determine significance at the second level using one-sample *t*-tests in a random-effects framework.

All first level models included the following regressors of no interest: 8 respiration and 6 movement regressors (with translation >1.5 mm or rotation >1° on any trial resulting in the inclusion of an additional outlier regressor), plus delta functions at the time of avoidance responses and shock receipt (avoidance response onsets included a parametric modulator representing reaction time, as overall we observed different mean RTs for GS and CS+ stimuli). In addition:

*Model 1: Expected value analysis.* We investigated encoding of modelled internal signals representing initial stimulus evaluation (i.e. the outcome that would be expected if no avoidance response was made), rather than values associated with chosen action on each trial (i.e. expected value of the outcome on that trial, following choice). For ease of interpretation, modelled internal value of not avoiding on a given trial ($V_{s,notAvoid,t}$) was multiplied by $-1$ to effectively represent predicted $P$ (shock) for that particular stimulus. The imaging model consisted of delta functions for CS onset (all trials), with parametric modulators of (i) estimated $P$(shock) according to the perceptual only model (ii) estimated $P$(shock) according to the perceptual +value based generalization model.

*Model 2: Prediction error analysis.* Prediction error (PE) was defined as the difference between predicted and actual outcome on a given trial, or ($R_t - V_{s,a,t}$). NB by definition this is equal to 0 on all trials where an avoidance response was made. The imaging model consisted of delta functions at the time of expected outcome delivery (all trials), with parametric modulators of (i) trial PE according to the perceptual only model (ii) trial PE according to the perceptual +value-based generalization model. Again, for ease of interpretability, PE terms were multiplied by $-1$ – such that positive PEs represented shock receipt (where predicted $P$(shock) was <1), and negative PEs represented shock omission (where predicted $P$(shock) was >0).

All regressors were convolved with a canonical haemodynamic response function, with correction for low-frequency drift using high-pass filtering (1/128 s) and correction for serially correlated errors by fitting of a first-order autoregressive process (AR(1)).

Computational model-based regressors were derived using individual subject free parameter values, and all regressors were orthogonalised during model estimation. SPM assigns variance to parametric modulators in a successive fashion, such that in an orthogonalised framework, a significant finding from a second parametric modulator represents that due to variance over and above that which has been assigned to the first modulator (**Mumford et al., 2015**). Due to the nature of our task design (i.e. that participants are only required to make motor responses on trials on which they wish to avoid that outcome of the presented stimulus), it is possible that expected value (predicted $P$(shock)) responses are partially contaminated by motor preparation responses (despite inclusion of appropriate nuisance regressors), due to the relative timing of these events. This should not be the case for the outcome prediction error analysis, as this focuses on trials where an avoidance response was not made (see Results). Additionally, there is greater variability in prediction error compared with expected value signals over the course of the task, making the former easier to discern statistically. However, changes in categorical stimulus representation associated with value are well evaluated using a multivariate approach (see below).

An initial cluster-forming threshold of p<0.001 (uncorrected), cluster size $\geq$10, was applied to 2nd level statistical maps, followed by cluster-level family wise error (FWE) rate correction at the whole-brain level ($p_{WB}$). Small-volume correction ($p_{SVC}$) was applied in *a priori* regions of interest (ROIs): namely the insula, amygdala, striatum, primary visual cortex (V1) and ventromedial prefrontal cortex (vmPFC) (see main text). ROIs were defined anatomically using the automatic anatomical labelling (aal) atlas (**Tzourio-Mazoyer et al., 2002**) in SPM ('striatum'=caudate + putamen+pallidum; 'V1'=Brodmann Area 17; 'vmPFC'=medial orbitofrontal cortices).

Only voxels present in all subjects were included in the analysis. For display purposes, statistical maps were thresholded at p<0.001 (uncorrected), and overlain on a high-quality mean MNI-space

structural image available as part of the MRIcroGL package. All quoted voxel coordinates refer to MNI space, in mm.

## Multivariate analysis

Representational similarity analysis (RSA) was carried out using materials from the RSA toolbox ([*Nili et al., 2014*]; available at github.com/rsagroup/rsatoolbox), run in Matlab.

For this analysis, time series data extracted from all voxels of each ROI were first multivariately noise normalized (data were beta images drawn from a simple categorical general linear model that consisted of stimulus onset by type and the same nuisance regressors as the univariate analyses). We calculated linear discriminant contrast values between pairs of stimulus categories (CS-, GS, CS+) as a robust estimate of representational dissimilarity (*Walther et al., 2016*). This approach involves construction of an optimal decision boundary (hyperplane) between pairs of multivariate representations (i.e. BOLD signal in all voxels, see *Figure 4a*). LDC values are a continuous measure of representational distance (dissimilarity) drawn by sampling of a dimension orthogonal to this decision boundary (Fisher's linear discriminant). To ensure distances were unbiased by noise (and therefore had a meaningful zero point), LDC values were estimated using a leave-one-out cross-validation approach across functional imaging runs (this constitutes a cross-validated estimate of the Mahalanobis distance; [*Walther et al., 2016*]).

*A priori* regions of interest were the same as for the univariate analysis. However, per our analysis plan, where possible anatomical ROIs were replaced by functional ROIs defined from the group-level univariate analysis. Specifically, the anterior insula and caudate clusters identified in *Figure 3b* were substituted for whole structure anatomical ROIs. This was done on the basis that (1) the univariate analysis indicated involvement of these voxels in specific value-related generalisation processes, and (2) previous analysis has shown that reliability of LDC RDMs falls off sharply for larger ROIs (>~250 voxels, [*Walther et al., 2016*]; anatomical ROIs for whole insula = 1019 voxels, for whole striatum = 3482 voxels; functional anterior insula ROI = 71 voxels, functional caudate ROI = 20 voxels, masks available at osf.io/25t3f).

## Questionnaire data

Questionnaire total and individual item scores were feature scaled (z-scored across participants) prior to further analysis.

Factor analysis was carried out as described in *Gillan et al (2016)*: implemented in R v3.4.0 (R Foundation for Statistical Computing), using the factanal function (psych package) with oblique (oblimin) rotation. The number of factors to extract was determined using the Cattell-Nelson-Gorsuch (*Gorsuch and Nelson, 1981*) method (nFactors package), whereby successive scree plot gradients are analysed to determine the 'elbow' point after which there is little gain in retaining additional factors. Factor names were chosen on the basis of the highest-loading items for each factor.

## Individual differences

Normality of distribution of individual variables (or within-subject differences in variables) was assessed using the Shapiro-Wilk test, and, where appropriate, non-parametric statistics were employed for pairwise tests.

In the fMRI sample, multivariate representational similarity estimates from all ROIs were compared to overall GS avoidance using an ordinary least squares multiple linear regression model. Mean avoidance across different trial types was z-scored within-participants, in order to gain a measure of *relative* GS avoidance (i.e. taking into individual variation in tendency to avoid on CS- and CS +trials).

Individual model parameters governing value-based generalization ($\sigma_A/\sigma_N$) were related to variables of interest (multivariate representational similarity in the fMRI sample, self-reported psychopathology in online sample) using weighted least squares multiple linear regression models. This method produces the maximum likelihood regression estimate when noise is not constant across measurements (i.e. data are heteroscedatic; [*Carroll and Ruppert, 1988*]). As the VBA toolbox yields the variance of posterior parameter estimates as well as the mean, weights were defined as the precision of individual parameter estimates (i.e. 1/posterior variance). Regression analyses were implemented in R using the function lm (psych package). Age (z-scored) and gender (binary scored as

male *vs* female/other) information were also included in all questionnaire data regression models as predictors of no interest. In R syntax: fit.wls = lm($\sigma_A$ ~ predictor(s)+ageZ + gender, weights = $\sigma_A$ precision)

Where candidate predictors were significantly collinear (as was the case for the questionnaire total scores data), they were implemented in separate regression models. Multiple comparisons correction for these models was achieved via the Nyholt-Bonferroni correction (*Li and Ji, 2005*), which yields a modified Bonferroni correction for non-independent (related) variables by estimating the 'effective number of independent variables' from the eigenvalues of their correlation matrix. Although we collected trait anxiety data in the MRI group (in order to characterise general anxiety levels in the sample, and screen out any individuals with undiagnosed pathologically significant anxiety), we did not plan to compare individual differences in behaviour to trait anxiety in this sample, as effect sizes from previous studies relating decision-making model parameters to psychological symptoms suggest this would be significantly underpowered (e.g. [*Gillan et al., 2016*]).

As a more robust test, we complemented our linear regression analyses with cross-validated regularized regression models, where all predictors were included in a single model. Specifically, we used least absolute shrinkage and selection operator (LASSO) regression (*Tibshirani, 1996*) with leave-one-out cross-validation. This approach effectively shrinks non-significant predictors to zero, and provides a more robust estimate of regression coefficients. This was implemented using the glmnet package in R. In R syntax: fit.cv = cv.glmnet(y= $\sigma_A$, x = all predictors, alpha = 1, nfolds = $N$, weights = $\sigma_A$ precision)

## Acknowledgements

This study was funded by the Wellcome Trust (grant number 097490/Z/11/A to BS). TWR was funded by a Wellcome Trust Senior Investigator Award (grant number 104631/Z/14/Z). The authors declare no relevant conflicts of interest.

## Additional information

### Funding

| Funder | Grant reference number | Author |
|---|---|---|
| Wellcome | 097490/Z/11/A | Ben Seymour |
| Wellcome | 104631/Z/14/Z | Trevor W Robbins |

The funders had no role in study design, data collection and interpretation, or the decision to submit the work for publication.

### Author contributions

Agnes Norbury, Conceptualization, Data curation, Software, Formal analysis, Investigation, Visualization, Methodology, Writing—original draft, Project administration, Writing—review and editing; Trevor W Robbins, Conceptualization, Methodology, Writing—review and editing; Ben Seymour, Conceptualization, Supervision, Funding acquisition, Methodology, Writing—review and editing

### Author ORCIDs

Agnes Norbury http://orcid.org/0000-0002-4377-3164
Ben Seymour http://orcid.org/0000-0003-1724-5832

### Ethics

Human subjects: Written, informed consent was obtained from all study volunteers. Both studies were approved by the University of Cambridge Psychology Research Ethics Committee (PRE.2015.101; PRE.2016.061).

### Decision letter and Author response

Decision letter https://doi.org/10.7554/eLife.34779.027

Author response https://doi.org/10.7554/eLife.34779.028

## Additional files

### Supplementary files

• Supplementary file 1. Demographic information for study participants. Unless otherwise specified, figures represent mean (SD). STAI, Spielberger State-Trait Anxiety Inventory (trait score only); AMI, Apathy Motivation Index; OCI-R, Obsessive-Compulsive Index (Revised); PHQ9, Physician's Health Questionnaire 9 (a brief measure of mood disorder symptoms); BIS-11, Barratt Impulsivity Scale (version 11); CSQ global, Cognitive Style Questionnaire (short-form) 'cognitive globalisation' subscale.
DOI: https://doi.org/10.7554/eLife.34779.016

• Supplementary file 2. Internal reliability of questionnaire scores in the AMT sample. STAI, Spielberger State-Trait Anxiety Inventory (trait score only); AMI, Apathy Motivation Index; OCI-R, Obsessive-Compulsive Index (Revised); PHQ9, Physician's Health Questionnaire 9 (a brief measure of mood disorder symptoms); BIS-11, Barratt Impulsivity Scale (version 11); CSQ, Cognitive Style Questionnaire (short-form).
DOI: https://doi.org/10.7554/eLife.34779.017

• Supplementary file 3. Individual item loadings derived from factor analysis of questionnaire data in the AMT sample. Item loadings are only shown above a threshold of ±0.25). Text in square brackets is to aid interpretation of reverse-scored items.
DOI: https://doi.org/10.7554/eLife.34779.018

• Transparent reporting form
DOI: https://doi.org/10.7554/eLife.34779.019

### Data availability

All relevant code for stimulus generation, data collection, and data analysis, in addition to raw behavioural data, is available at the project's Open Science Framework page (osf.io/25t3f). Raw functional imaging data is deposited at openfMRI (openfmri.org/dataset/ds000249) and derived statistical maps are available at NeuroVault (neurovault.org/collections/3177).

The following datasets were generated:

| Author(s) | Year | Dataset title | Dataset URL | Database, license, and accessibility information |
|---|---|---|---|---|
| Agnes Norbury | 2017 | Value generalization in human avoidance learning | https://neurovault.org/collections/3177/ | Publicly available at Neurovault (accession no. 3177) |
| Agnes Norbury | 2017 | Value generalization in human avoidance learning | https://osf.io/25t3f/ | Publicly available at the Open Science Framework |
| Agnes Norbury, Ben Seymour | 2018 | Value generalization in human avoidance learning | https://openneuro.org/datasets/ds000249/versions/00002 | Publicly available at Open Neuro |

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
