## [Decision Letter]

Thank you for submitting your article "Value generalization in human avoidance learning" for consideration by *eLife*. Your article has been favorably evaluated by Sabine Kastner (Senior Editor) and three reviewers, one of whom is a member of our Board of Reviewing Editors. The reviewers have opted to remain anonymous.

The reviewers have discussed the reviews with one another and the Reviewing Editor has drafted this decision to help you prepare a revised submission.

Summary:

This is an interesting and compelling investigation on generalization processes during human avoidance learning. The authors have used a clever experimental paradigm to test whether the generalization during avoidance learning can be decomposed into the processes related to perceptual and value-related processes, and to examine how such different components of generalization map on to different neural substrates. They found that a model with perceptual, aversive, neutral components of generalization accounted for the observed behavioral data best, and that signals related to the aversive and neutral components of value learning were identified in different brain areas. Finally, the aversive component of the generalization was related to the anxiety-related traits.

Essential revisions:

The reviewers have raised two major concerns that must be addressed appropriately before the paper can be accepted. This might require some additional analyses and discussion of the data.

1) The experimental task confounds shock expectations with motor responses. This makes it difficult to interpret value-related fMRI activity. In particular, because motor responses were only made on a subset of trials (those with high probability of being paired with shock), it is unclear how signals related to shock expectation can be dissociated from those related to motor preparation/execution. Parametric value estimates from a model that predicts behavior well will necessarily also correlate with motor-related fMRI signals. One way to overcome this problem would be to estimate parametric value-bases responses separately for trials with and without avoidance responses. Alternatively, the GLM should include a parametric regressor (as first regressor) which codes for whether or not an avoidance response was made.

2) It is not entirely clear whether the current task measures classical generalization-like behavior. Peak shifts are considered to be an indicator of generalization. In the current task, one would expect peak shifts for the GS's corresponding to one of the two CS+ (the spiky GS closer to the CS – in Figure 1B). However, peak shifts are not observed in the fMRI sample. This is inconsistent with classical indicators of generalization, raising the question what this task really measures. Typical generalization experiments probe behavior in response to a large number of GS. This makes it virtually impossible for participants to directly acquire stimulus-outcome associations for each and every GS, and subjects have to rely on generalization to respond optimally. In contrast, the current task uses only two GS's, potentially allowing subjects to directly learn GS-outcome associations. Thus, participants (at least in the fMRI sample) might have approached this task as a perceptual challenging but otherwise normal learning task. The implication of this limitation should be discussed.

---

## [Author Response]

Essential revisions:The reviewers have raised two major concerns that must be addressed appropriately before the paper can be accepted. This might require some additional analyses and discussion of the data.1) The experimental task confounds shock expectations with motor responses. This makes it difficult to interpret value-related fMRI activity. In particular, because motor responses were only made on a subset of trials (those with high probability of being paired with shock), it is unclear how signals related to shock expectation can be dissociated from those related to motor preparation/execution. Parametric value estimates from a model that predicts behavior well will necessarily also correlate with motor-related fMRI signals. One way to overcome this problem would be to estimate parametric value-bases responses separately for trials with and without avoidance responses. Alternatively, the GLM should include a parametric regressor (as first regressor) which codes for whether or not an avoidance response was made.

We took the decision that participants should only be required to make motor responses on avoidance trials in an attempt to increase the ecological validity of our paradigm as a probe of avoidance behaviour. Specifically, we wanted to frame the decision to be made on each trial as one to stay and sample a particular stimulus, versus deciding to withdraw (avoid the outcome of that stimulus) – thereby incurring a small cost (analogous to time and/or energy cost in the real world). This decision was made on the basis that framing two choice decisions in this way (versus e.g. an approach-avoidance framework, where selecting either option requires a behavioural response) can affect the choices participants make (e.g. Wright et al., 2013, Frontiers in Neuroscience).

However, we agree that this creates an issue for certain aspects of MRI analysis – specifically our expected value analysis as modelled internal probability of shock (our expected value regressor) incurs collinearity with the decision to avoid. In order to address this issue, we have re-run all our univariate models with an additional nuisance regressor representing avoidance responses (delta function at time the response was made), parametrically modulated by reaction time (time from stimulus onset to avoidance) – as we observed a difference in mean reaction time between CS+ and GS trials (see Results and newFigure 3). The inclusion of this regressor in our re-analysis should significantly ameliorate concerns over non value-related motor confound effects (i.e. regress out effect of makingvs. not making a button press on a given trial). It is not possible under our design to disambiguate motor preparation responses that may covary with value (e.g. relating to movement force/vigour) – however, as long as this relates to response generation, rather than other potential corollaries such as sensory feedback, it should be considered a part of the generalization response itself, rather than representing a confound (for example, almost all human decision-making studies involve a motor response of some sort).

The core results of both the expected value and outcome prediction error analyses were unaltered under the new model specification – specifically, we still found evidence of univariate encoding of variance specific to regressors derived from a value-based generalization model in the anterior insula and dorsal striatum (new Figure 3). Indeed, removing the motor preparation confound improved the result on the outcome prediction error analysis, such that significant clusters were now observed in the dorsal striatum (bilateral putamen and right pallidum – see new Figure 3C).

As well as updating the Results and relevant figures, we have edited the Materials and methods to make it clearer exactly which nuisance regressors were included in all models (please see below). We also updated the simple categorical model that the multivariate data was drawn from to include avoidance response regressors (modulated by reaction time) – please see Results and new Figure 4. Finally, we have added a note to the Materials and methods to explain that, despite inclusion of appropriate motor response regressors, there is still a chance that the expected value analysis may be contaminated by motor preparation responses, due to the relative timing of trial events (please see below).

“All first level models included the following regressors of no interest: 8 respiration and 6 movement regressors (with translation >1.5mm or rotation >1° on any trial resulting in the inclusion of an additional outlier regressor), plus delta functions at the time of avoidance responses and shock receipt (avoidance response onsets included a parametric modulator representing reaction time, as overall we observed different mean RTs for GS and CS+ stimuli).”

“Representational similarity analysis (RSA) was carried out using materials from the RSA toolbox (Nili et al., 2014; available at github.com/rsagroup/rsatoolbox), run in Matlab. For this analysis, time series data extracted from all voxels of each ROI were first multivariately noise normalized (data were beta images drawn from a simple categorical general linear model that consisted of stimulus onset by type and the same nuisance regressors as the univariate analyses).”

“Due to the nature of our task design (i.e., that participants are only required to make motor responses on trials on which they wish to avoid that outcome of the presented stimulus), it is possible that expected value (predicted *P*(shock)) responses are partially contaminated by motor preparation responses (despite inclusion of appropriate nuisance regressors), due to the relative timing of these events. This should not be the case for the outcome prediction error analysis, as this focuses on trials where an avoidance response was not made (see Results).”

2) It is not entirely clear whether the current task measures classical generalization-like behavior. Peak shifts are considered to be an indicator of generalization. In the current task, one would expect peak shifts for the GS's corresponding to one of the two CS+ (the spiky GS closer to the CS – in Figure 1B). However, peak shifts are not observed in the fMRI sample. This is inconsistent with classical indicators of generalization, raising the question what this task really measures. Typical generalization experiments probe behavior in response to a large number of GS. This makes it virtually impossible for participants to directly acquire stimulus-outcome associations for each and every GS, and subjects have to rely on generalization to respond optimally. In contrast, the current task uses only two GS's, potentially allowing subjects to directly learn GS-outcome associations. Thus, participants (at least in the fMRI sample) might have approached this task as a perceptual challenging but otherwise normal learning task. The implication of this limitation should be discussed.

We would argue that it is not necessary to use multiple generalization stimuli probes around a particular CS to demonstrate generalization, as long as you can demonstrate that the probe stimuli you have used are reliably perceptually distinguishable from that CS, and you make not particular claims about the precise nature of the generalization function. Assuming this function is constant across individuals, rank ordering of individual function width estimates should be fairly stable, regardless of the exact function chosen (e.g., both exponential and Gaussian functions have been proposed in the past, and indeed may both fit equally well a small number of noisy datapoints derived from human physiological data). Under these circumstances, use of single probe GSs around CSs is the most statistically efficient design – particularly under circumstances such as fMRI where individual trials are necessarily quite long.

We did not observe a peak shift effect in overall proportionate choice data in our MRI sample – however this is also the case for other recent human studies of aversive generalization using multiple GS stimuli and similar sample sizes (Onat and Büchel 2015; Laufer, Israeli and Paz, 2016), so we would argue this is not a pre-requisite for inferring any kind of generalization process is at play. Interestingly, peak shift effects have previously primarily been demonstrated using reward-based paradigms – it could be that aversively-conditioned peak shift is a less robust phenomenon, and we are simply underpowered to detect it at standard functional imaging sample sizes (cf our significantly larger online cohort).

The reviewer raises the important point of considering how participants conceptualise our task, given we do not use multiple GSs, and therefore explicitly (or implicitly) frame the experiment as a generalization problem. Under these circumstances, is it possible that our participants are simply solving the task by considering each stimulus as an independent learning target and not using generalization at all? Using our computational framework, it is possible for us to test this hypothesis explicitly, by comparing models of the task where participants transferred value between states (stimuli) on the basis of their perceptual similarity (i.e., generalized), to a model where all states are learned about independently (with no supra-perceptual threshold transfer of value). Indeed, this is the exact analysis illustrated in Figure 2A. Bayesian model comparison revealed that in neither cohort was it likely the participants were using model where task stimuli were learnt about independently (‘perceptual only model’ in Figure 2) – either at the group level (estimated probability that this model was the true population model was ~0 in both groups), or in a subset of individuals (proportion of participants for whom this was the most likely model was <0.1 in the MRI group, and ~0 in the online group). Further, it is possible that lack of explicit or implicit framing of the task as using a generalization design guarded against the influence of demand characteristics in participants’ responding (particularly e.g. in subjective ratings data).

We have added a brief discussion of this design issue to the Materials and methods (please see below), and attempted to make the motivation underlying the model comparison analysis clearer in the Results section.

“On a low frequency of trials, shapes were generalization stimuli (GSs; 2 presentations of each GS per 38 trial block). […] Frequency of individual GS presentation (10 per GS) was comparable to recent functional imaging studies of Pavlovian generalization (e.g. 7 and 34 presentations per GS during generalization test phases, respectively: Laufer et al., 2016; Onat and Büchel, 2015).”